# BEND: Benchmarking DNA Language Models on Biologically Meaningful Tasks

**Frederikke I. Marin**[1,2]*, **Felix Teufel**[3,4]*, **Marc Horlacher**[5], **Dennis Madsen**[3],
**Dennis Pultz**[1], **Ole Winther**[4,6], **Wouter Boomsma**[2]
[1]Bioinformatics & Design, Enzyme Research, Novozymes A/S
[2]Department of Computer Science, University of Copenhagen
[3]Digital Science & Innovation, Novo Nordisk A/S
[4]Department of Biology, University of Copenhagen
[5]Computational Health Center, Helmholtz Center Munich
[6]DTU Compute, Technical University of Denmark

## Abstract

The genome sequence contains the blueprint for governing cellular processes. While the availability of genomes has vastly increased over the last decades, experimental annotation of the various functional, non-coding and regulatory elements encoded in the DNA sequence remains both expensive and challenging. This has sparked interest in unsupervised language modeling of genomic DNA, a paradigm that has seen great success for protein sequence data. Although various DNA language models have been proposed, evaluation tasks often differ between individual works, and might not fully recapitulate the fundamental challenges of genome annotation, including the length, scale and sparsity of the data. In this study, we introduce **BEND**, a **Ben**chmark for **D**NA language models, featuring a collection of realistic and biologically meaningful downstream tasks defined on the human genome. We find that embeddings from current DNA LMs can approach performance of expert methods on some tasks, but only capture limited information about long-range features. BEND is available at `https://github.com/frederikkemarin/BEND`.

## 1 Introduction

Within the last two decades, the cost of sequencing whole genomes has significantly decreased, having led to an extraordinary wealth of genomic DNA sequences. This has improved our understanding of genetic variation among human genomes and introduced genomes of hitherto understudied species. However, the generation of experimental data to annotate and understand these genomic sequences has not kept pace.

At the same time, Natural Language Processing (NLP) has demonstrated the power of large-scale models to capture signals in sequences by masking and reconstructing them in a self-supervised manner. The success of masked language modeling (MLM) has extended to the biological domain Rao et al. (2019); Bepler & Berger (2021); Madani et al. (2023); Rives et al. (2019), with protein language models (pLMs) now being widely used for prediction tasks on protein sequences. The availability of unlabeled genomic sequences and, in many organisms, limited labeled data appear to make language modeling a natural fit for DNA. DNA language models (LMs) have indeed started to emerge, but while the paradigms of NLP have been easy to transfer to proteins, the same may not be true for modeling genomes, as they present unique challenges: signals can have an extremely long length range, high-signal regions are sparse, and even in those regions the density of signal is lower compared to proteins.

In this paper, we present BEND, a **Ben**chmark for **D**NA Language Models, a collection of realistic and biologically meaningful downstream tasks. BEND aims to provide a standardized set of tasks that measure the ability of LMs to capture the intricacies of genomic data, and to help advance this nascent field. In summary, BEND contributes:

---

*Equal contribution

- **Seven curated tasks and datasets**, probing understanding of different DNA functional elements over a variety of length scales.

- **Experiments covering DNA LMs from six different sources.** To our knowledge, this represents first evaluation of all publicly available self-supervised DNA LMs suitable for the human genome together with appropriate baseline methods.

- **An adaptable benchmarking framework** for preparing embeddings and training lightweight supervised models.

- **Result: DNA LMs approach expert method performance on some tasks**. However, no LM consistently outperforms all others, and reasoning over very long contexts, as e.g. required for finding enhancers, is still challenging.

- **Result: DNA LMs can learn distinct features in masked language modeling.** Some LMs' embeddings primarily capture information about gene structure, while others focus on noncoding regions.

## 2 BACKGROUND

### 2.1 EUKARYOTIC DNA ORGANIZATION AND TERMINOLOGY

In order to facilitate understanding how different prediction tasks relate to various aspects of the genome, we briefly discuss the fundamental structure and function of eukaryotic genomic DNA (Figure 1). DNA is a linear polymer of four nucleotide bases, which are represented by the four letters A, C, G and T. It consists of two complementary *strands* that form a *double helix* by *base pairing* the bases A, T, and C, G respectively.

Genomic DNA is physically organized in a hierarchical manner. The DNA polymer is coiled around *histone* proteins, which reduces its physical length and plays a role in regulation. A complex of 8 histone proteins together with coiled DNA is called a *nucleosome*. Nucleosomes further condense to form *chromatin* fibers, which occur in compact (closed) or loose (open) form. This controls the accessibility of the DNA sequence to the transcriptional machinery, a process tightly regulated by chemical modifications of the histones (Bannister & Kouzarides, 2011). Chromatin can form loops, which allows regions distant in the sequence to be close in physical space. DNA appears in independent modules called *chromosomes*, which are typically millions of base pairs (bp) in length.

The genome contains *genes*, segments that are transcribed to RNA molecules and potentially translated to proteins. Protein-coding genes are structured as *introns* and *exons*. For expression, a gene is first transcribed to a pre-mRNA molecule, and introns are removed via *splicing*. This combines the exons to one contiguous sequence that encodes the protein. Flanking nucleotides in the RNA that do not code for the protein are called untranslated regions (UTRs) and can have regulatory function. In addition, genes are associated with regulatory regions such as *promoters*, *enhancers*, *silencers* and *insulators* that modulate their expression. Some elements, such as promoters, may lie in close proximity to the start of the gene, the *transcription start site* (TSS). Others can appear multiple thousands bp away from the gene, but mediate their effect by physical proximity.

### 2.2 LANGUAGE MODELING FOR BIOMOLECULAR SEQUENCES: FROM PROTEINS TO DNA

Over the last years, language modeling has achieved breakthroughs in representation learning for protein property and structure prediction, with transformer-based pLMs emerging as powerful foundation models, capable of learning long-range interactions fully unsupervised (Rives et al., 2019; Elnaggar et al., 2022; Lin et al., 2023). The development of pLMs benefitted from the availability of standardized, representative benchmarks, such as TAPE (Rao et al., 2019) and PEER (Xu et al., 2022), as well as long-running protein machine learning tasks with an emphasis on fair benchmarking to measure progress (Kryshtafovych et al., 2021; Zhou et al., 2019).

While LMs have been extremely successful for modeling proteins, key differences between the two types of macromolecules hinder their widespread adoption for DNA. A typical protein consists of 400-500 amino acids, which are represented as tokens from an alphabet of size 20. The analogy of amino acid tokens with word tokens in NLP, as well as the fact that size of inputs to pLMs and NLP models are on the same order of magnitude, made methods developed for NLP directly transferable to protein data, with little to no methodological adaption required (Rao et al., 2020; Elnaggar et al., 2022). The alphabet of DNA is significantly smaller (4 tokens), while at the same time sequences,

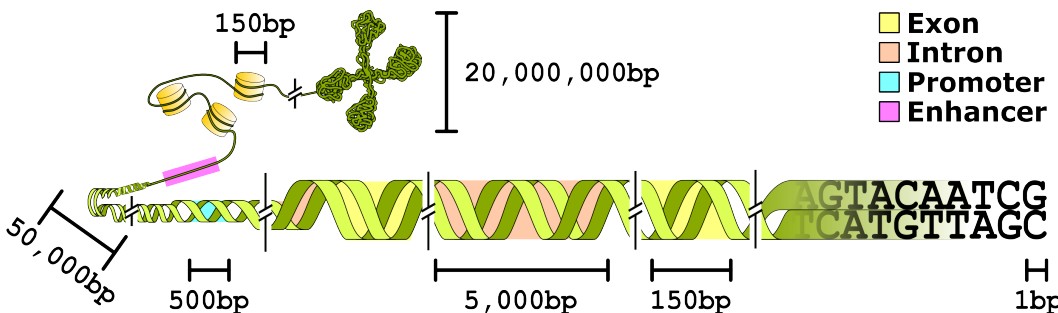

**Figure 1:** The organization of eukaryotic genomic DNA. Numbers are indicative examples for the human genome. Genes are structured as alternating introns (average: 5,400 bp) and exons (average: 170 bp), and have a promoter regulatory element before their TSS. Enhancers can be thousands of bp away from the gene. DNA is wrapped around histone proteins and densely packed as a chromosome.

such as those of genes, are considerably longer and have no naturally defined border, as e.g. the position of the most distant relevant regulatory element is typically unknown. In contrast, protein sequences are naturally self-contained and, being the final gene product, have a significantly higher information density. Together, sparsity and long sequences pose unique challenges to DNA LMs.

## 2.3 RELATED WORKS

### 2.3.1 DNA LANGUAGE MODELS

The first available DNA LM was DNABERT (Ji et al., 2021), a 12-layer BERT (Devlin et al., 2018) model trained on sequences of length 512 from the human genome. Sequences were tokenized as k-mers using a sliding window. DNABERT was evaluated by fine-tuning on tasks comprising promoter, transcription factor (TF) binding site and splice site (SS) prediction.

A growing number of DNA LMs has been proposed since the release of DNABERT. These include the Genomic Pretrained Network (GPN) (Benegas et al., 2023), FloraBERT (Levy et al., 2022), the Nucleotide Transformer (NT) (Dalla-Torre et al., 2023), Species-aware LM (Gankin et al., 2023), GENA-LM (Fishman et al., 2023), DNABERT-2 (Zhou et al., 2023) and HyenaDNA (Poli et al., 2023). With the exception of HyenaDNA, models were trained using the MLM objective, but differ in their model architectures, tokenization strategies and training data.

GPN uses dilated convolution layers rather than a transformer model. It showed strong performance for zero-shot prediction of variant effects in the *A. thaliana* genome it was trained on. Qualitative results showed that GPN captures information about gene structure and motifs of binding sites.

Nucleotide Transformer introduced the first large-scale transformer-based DNA LMs. All NT models share the same architecture, but differ in their number of training genomes and model parameters. Models were trained on either the human reference genome, 3,202 different genetically diverse human genomes or a selection of 850 genomes from a range of species. To increase the receptive field of the model, sequences were tokenized as 6-mers, allowing for processing sequences of up to 5,994 bp in length. A second generation of multispecies models released later (NT-V2) extended the input length to 12,282 bp. The NT models were evaluated on tasks comprising promoter, SS, histone modification and enhancer prediction with a context length of up to 600 bp.

GENA-LM (Fishman et al., 2023) proposed multiple medium-size LMs trained on human and multi-species genomes based on BERT and the BigBird (Zaheer et al., 2020) architecture for long sequences. Byte-Pair Encoding (BPE) was used for tokenization to further increase the receptive field, enabling an input length of about 36,000 bp. Models were evaluated on tasks comprising promoter, SS, enhancer, chromatin profile and polyadenylation site prediction. While covering the same biological phenomena, tasks were defined differently than in NT. Similarly, DNABERT-2 (Zhou et al., 2023) replaced DNABERT's k-mer tokenizer with BPE and pre-trained on multi-species genomes, while GROVER (Sanabria et al., 2023) adopted BPE for the human genome.

Predating NT and GENA-LM, FloraBERT (Levy et al., 2022) proposed pre-training on 93 different plant genomes to enable transfer learning for predicting gene expression. However, FloraBERT was

trained exclusively on promoter-containing sequences. As this requires features to already be annotated in the genome, it can be considered a departure from the paradigm of fully self-supervised learning. Similarly, Gankin et al. (2023) pre-trained on 3' UTRs from 1,500 fungal genomes. Species information was made explicit by providing a species label with each sequence to the model.

HyenaDNA (Nguyen et al., 2023) introduced a collection of autoregressive LMs, trained using the next token prediction objective at single-nucleotide resolution on the human genome. The Hyena LM architecture (Poli et al., 2023) enabled scaling to input lengths of up to 1 million nucleotides. HyenaDNA models were evaluated by fine-tuning on NT's supervised tasks and the Genomic Benchmarks (Grešová et al., 2023) collection, outperforming NT on the majority of tasks.

A number of DNA LMs were proposed without making trained models available. These comprise the original BigBird (Zaheer et al., 2020), GeneBERT, which includes the prediction of ATAC-seq signals in the pre-training stage, MoDNA (An et al., 2022), with a motif prediction task as an additional objective, the BERT-based LOGO (Yang et al., 2021), and Revolution (Cheng et al., 2023), which adopts convolutions with circular padding.

### 2.3.2 SUPERVISED LEARNING ON DNA

Developing models on genomic DNA sequences for the prediction of properties and understanding of transcriptional regulation has long been a central task of computational genomics research. The availability of large-scale functional genomics data and advancements in deep learning techniques have brought progress in predicting various genomic features directly from DNA sequences. DeepBind (Alipanahi et al., 2015) and DeepSEA (Zhou & Troyanskaya, 2015) were two of the first methods leveraging shallow CNNs for predicting TF binding and chromatin features, respectively. DeepCpG (Angermueller et al., 2017) predicts DNA methylation via a CNN/GRU architecture. Basset (Kelley et al., 2016) and ChromTransfer (Salvatore et al., 2023) model chromatin state in a cell type specific manner by predicting the presence of DNase-I peaks. Using chromatin state as an auxiliary input, DeepChrome (Singh et al., 2016) predicts gene expression via multi-modal learning on DNA sequence and histone mark information.

Recently, methods for predicting gene expression have leveraged information across thousands of functional genomic tracks by training in a large-scale, multi-task fashion. Basenji (Kelley et al., 2018) and Enformer (Avsec et al., 2021) demonstrated state-of-the-art performance for gene expression prediction from DNA sequence alone, by integrating genomic information across up to 200 kilobases and multi-task training across several genome-wide functional tasks, including DNase-I activity and CAGE signal prediction. Similarly, Sei (Chen et al., 2022) models cis-regulatory TF binding, chromatin accessibility and histone modification profiles across a large range of cell types.

### 2.3.3 BENCHMARK COLLECTIONS ON DNA

Genomic Benchmarks (Grešová et al., 2023) features balanced classification tasks on DNA sequences with median lengths ranging from 200 to 2,381 bp. The benchmark covers the classification of functional elements and the prediction of a sequence's origin. The element classification tasks are defined on human DNA, with one task covering *D. melanogaster* additionally. For each task, only performance of a baseline supervised neural network model was reported.

DNABERT-2 introduced Genome Understanding Evaluation (GUE), a collection of classification tasks ranging from 70 to 1,000 bp. On the human genome, it includes classification of promoter, SS and TF binding sequences. It covers other species with a TF binding task on mouse, a histone modification task on yeast and a Covid variant classification task on viruses. DNABERT, DNABERT-2 and NT were evaluated. No non-LM task-specific baselines are included in GUE.

The authors of NT provide a public leaderboard for their tasks, comprising promoter (human/mouse), enhancer (human), SS (human/multispecies) and histone modification (yeast) prediction with lengths ranging from 300 to 600 bp. NT is compared to DNABERT, DNABERT-2, HyenaDNA and Enformer. No task-specific baselines are included in the leaderboard.

### 2.3.4 MOTIVATION OF BEND

While existing DNA LMs have reported good performance on the tasks on which they were evaluated, evaluation strategies to date have shown limited consistency across individual works, with GUE constituting the most recent attempt at benchmarking on equal terms. Beyond comparability, it is important to ensure that benchmark tasks reflect the complexity and characteristics of real-world genome analysis. In practice, genomes are vast, and functional regions are sparsely distributed

**Table 1:** Overview of the tasks included in the benchmark. Nucleotide-wise tasks require the prediction of a sequence of labels with the same length as the input. In sequence-wise tasks the whole input sequence is to be classified. In binned tasks, multiple nucleotides share a label.

| Task | Type (# labels) | # Samples | Length range | Evaluation (# train/val/test) | Metric | Source |
|------|------|------|------|------|------|------|
| Gene finding | Nucleotide-wise Multiclass (9) | 5,976 | 1,433 - 14,000 bp | 4780/597/597 | MCC | GENCODE (Frankish et al., 2021) |
| Enhancer annotation | Binned (128bp) Binary | 285 | 100,096 bp | 10-fold CV | AUPRC | Fulco et al. (2019), Gasperini et al. (2019), Enformer (Avsec et al., 2021) |
| Chromatin accessibility | Sequence-wise Multilabel (125) | 2,005,617 | 512 bp | 1,354,042/ 279,422/372,153 | AUROC | ENCODE Project Consortium (2012) |
| Histone modification | Sequence-wise Multilabel (18) | 612,081 | 512 bp | 420,713/ 70,801/120,567 | AUROC | ENCODE Project Consortium (2012) |
| CpG methylation | Sequence-wise Multilabel (7) | 959,039 | 512 bp | 743,095/ 109,717/106,227 | AUROC | ENCODE Project Consortium (2012) |
| Noncoding variant effects (expression) | Sequence-wise Binary | 105,263 | 512 bp | zero-shot | AUROC | DeepSEA (Zhou & Troyanskaya, 2015) |
| Noncoding variant effects (disease) | Sequence-wise Binary | 295,495 | 512 bp | zero-shot | AUROC | ClinVar (Landrum et al., 2020) |

throughout the genome. While there are tasks on DNA that are inherently local, such as classifying functional regions (e.g. classifying TF binding sites), it needs to be recognized that such tasks do not allow us to evaluate a model's understanding of the genome over longer ranges.

Therefore, focusing solely on tasks on short sequences, such as distinguishing promoter from non-promoter sequences, falls short of evaluating the extent to which a model's representations capture complex features of genomic organization, preventing us from measuring benefits of modeling the genome with larger context windows. For instance, reporting performance on predicting SSs, which can be done on short sequences, does not allow us to evaluate how useful a model would be for gene finding over longer ranges, a common task in genome annotation.

To provide a more comprehensive assessment, BEND proposes genomic tasks that rely less on prior knowledge of feature positions and require reasoning over potentially long contexts. The tasks cover a range of length scales, selected to be both biologically relevant and to cover a variety of DNA properties. The tasks explore representations at different resolutions, requiring modelling of DNA at single bp resolution as well as over longer stretches (Table 1). We establish our benchmark on the human genome, as it offers ample experimental data for the derivation of tasks, has a complex organization, and was the focus of most published DNA LMs.

## 3 TASKS AND DATASETS

We introduce the collection of tasks included in BEND. For each task, we additionally provide a datasheet (Gebru et al., 2018) in section A.1. All tasks are provided in `bed` format, listing the genome coordinates of samples (A.2). This makes it convenient to include more flanking context without reprocessing the data, should future works find it useful to take more bp into account.

### 3.1 GENE FINDING

**Definition** Gene finding is a multiclass problem where each nucleotide is either part of an exon ($E_{F/R}$), intron ($I_{F/R}$), a donor ($D_{F/R}$) or acceptor ($A_{F/R}$) splice site or a noncoding region ($NC$). The $F/R$ subscript denotes whether the gene is located on the forward or reverse strand.

**Biological relevance** Annotating genes and identifying coding sequences is a key step in genome annotation and protein discovery. It requires a model to use local context to identify correct reading frames and codon structure, while using longer range signals to propagate the location of SS to distant bp between SS, and correctly annotate them as lying in introns or exons. Introns vary in length from a few hundred to several thousand bp, requiring an LM to understand long-range dependencies.

**Data** GENCODE (Frankish et al., 2021) gene annotations were processed to construct sequences of nucleotide labels $y \in \{E_F, D_F, I_F, A_F, E_R, D_R, I_R, A_R, NC\}$ for each gene. Detailed processing is laid out in A.1.1. Samples were partitioned at 80% identity following AUGUSTUS' recommendations (Stanke & Waack, 2003). It should be noted that there is a large label imbalance as there is only one donor and acceptor site per intron segment.

**Metric** We compute the multi-class Matthews correlation coefficient (MCC) (Gorodkin, 2004) over all bp. The MCC is used as it is robust to the inherently highly uneven label ratios of this task.

## 3.2 ENHANCER ANNOTATION

**Definition** Enhancer annotation is the problem of finding enhancer regions for a given gene. We define enhancer annotation as a binary classification task. Given a sequence of gene-adjacent genomic DNA that contains enhancers, a binary label indicating whether it is part of an enhancer needs to be predicted for each segment of 128bp.

**Biological relevance** Enhancers are short, noncoding segments that contribute to regulating gene expression. They can be located anywhere from a few thousand to a million bp away from their target gene and work by being brought into physical proximity to the gene's promoter. Their annotation is a highly challenging task that requires detection of long-range interactions.

**Data** Experimentally validated enhancer-gene pairs were taken from CRISPR interference experiments (Fulco et al. (2019); Gasperini et al. (2019) and paired with the main TSS of each gene from Avsec et al. (2021). We extracted a sequence of 100,096 bp centered on the TSS for each gene. Each 128bp were annotated with a binary label $y \in \{0, 1\}$ indicating whether the bin is part of an enhancer, yielding a label sequence of length 782. Detailed processing is laid out in A.1.2. Samples were partitioned based on chromosomes.

**Metric** The AUPRC is computed over all labels. As the number of samples is too limited for measuring performance robustly on a single test split, we perform 10-fold cross-validation in order to evaluate performance over all samples.

## 3.3 CHROMATIN ACCESSIBILITY PREDICTION

**Definition** Chromatin accessibility prediction is a multilabel task where sequences are classified as being in open or closed chromatin across a range of cell types.

**Biological relevance** Dynamically modulating chromatin accessibility is a key mechanism for the cell type specific regulation of gene expression, as binding of the transcription machinery is highly dependent on the accessibility of DNA elements, including promoters, enhancers and TSS.

**Data** DNase I hypersensitive sites were obtained from ENCODE (ENCODE Project Consortium, 2012) for 125 cell types. Following the preprocessing of Kelley et al. (2016), segments of length 512 bp were labeled with binary vectors $\mathbf{y} \in \{0, 1\}^{125}$, with $y_i = 1$ if the chromatin is open for the $i$'th cell type. Detailed processing is laid out in A.1.3. Samples were partitioned based on chromosomes.

**Metric** The AUROC is computed for each label and averaged.

## 3.4 HISTONE MODIFICATION PREDICTION

**Definition** Histone modification prediction is a multilabel task, where the histones which are part of the nucleosomes of a given DNA sequence are labeled with one or more histone marks.

**Biological relevance** Histone proteins are key to the organisation of DNA into chromatin. Modifications of histones modulate chromatin structure and thus contribute to regulating chromatin accessibility and gene expression. Histone modification prediction requires modeling local binding of TFs as well as long-range regulation, such as by distant enhancers.

**Data** Histone ChIP-seq data for 11 histone marks and 18 replicates in the K562 cell line was obtained from ENCODE. Detailed processing is laid out in A.1.4 and follows the methodology of 3.3. Each sample is a sequence of length 512 bp with a label vector $\mathbf{y} \in \{0, 1\}^{18}$, such that $y_i = 1$ if a histone bound to this sequence carries mark $i$. Samples were partitioned based on chromosomes.

**Metric** The AUROC is computed for each label and averaged.

## 3.5 CpG METHYLATION PREDICTION

**Definition** CpG methylation prediction is a multilabel classification task, where a given CpG site is either methylated or unmethylated in different cell lines.

**Biological relevance** Methylation of cytosine nucleotides in CpG sites is a prominent form of epigenetic modification and plays a key role in the repression of gene expression.

**Data** Bisulfite sequencing data for 7 human cell lines was obtained from ENCODE. Detailed processing is laid out in A.1.5. Each sample is a sequence of length 512 bp centered on the CpG site with a label vector $\mathbf{y} \in \{0, 1\}^7$, such that $y_i = 1$ if the C is methylated. Samples were partitioned based on chromosomes.

**Metric** The AUROC is computed for each label and averaged.

## 3.6 NONCODING VARIANT EFFECTS (EXPRESSION AND DISEASE)

**Definition** Predicting variant effects is a binary problem, where single-bp mutations are classified as either having an effect or not. We treat classification as a zero-shot task, using the cosine distance

**Table 2:** Overview of the LMs applicable to the human genome included in the benchmark.

| Model | Seq length | Trained on | Architecture | Source |
|---|---|---|---|---|
| AWD-LSTM | Infinite[a] | Multispecies | RNN | This work |
| Dilated ResNet | 10,000 | Human Ref[d] | CNN | This work |
| Nucleotide Transformer | 5,994 | Human Ref[d] | BERT | Dalla-Torre et al. (2023) |
| Nucleotide Transformer | 5,994 | Multispecies | BERT | Dalla-Torre et al. (2023) |
| Nucleotide Transformer | 5,994 | 1000 Genomes project[d] | BERT | Dalla-Torre et al. (2023) |
| Nucleotide Transformer V2 | 12,282 | Multispecies | BERT | Dalla-Torre et al. (2023) |
| DNABERT | 512 | Human Ref[d] | BERT | Ji et al. (2021) |
| DNABERT-2 | Infinite[b] | Multispecies | BERT | Zhou et al. (2023) |
| GENA-LM | 4,500 | 1000 Genomes project[e] | BERT | Fishman et al. (2023) |
| GENA-LM | 36,000 | 1000 Genomes project[e] | BigBird | Fishman et al. (2023) |
| HyenaDNA | 1,000,000 | Human Ref[d] | Hyena | Nguyen et al. (2023) |
| HyenaDNA | 1,000 | Human Ref[d] | Hyena | Nguyen et al. (2023) |
| GROVER | 8,160[c] | Human Ref[d] | BERT | Sanabria et al. (2023) |

[a] As the LSTM compresses all preceding tokens into a single hidden state, it can technically process infinite sequences, even though it was trained at finite lengths and might not have learnt to exploit such long contexts.
[b] DNABERT-2 uses ALiBi (Press et al., 2022) to encode position, which can technically scale to any sequence length. In practice, the model was trained on finite lengths and the authors recommend embedding sequences below 10,000 bp.
[c] No explicit length was reported in bp. The indicated number was derived by considering 510 BPE tokens of size 16.
[d] Schneider et al. (2017), [e] McVean et al. (2012)

in embedding space between a variant nucleotide and its reference nucleotide as the prediction score.
**Biological relevance**     Single-bp variants in noncoding regions can have functional consequences by altering gene expression levels or causing disease. This task probes the LM's understanding of local context and potentially the structure of regulatory motifs. We focus on noncoding regions, as coding variant effects can be predicted with high accuracy by modeling the mutation in the resulting protein sequence (Frazer et al., 2021).
**Data**     For expression variants, we adapt the DeepSEA dataset (Zhou & Troyanskaya, 2015). For disease-associated variants, we process ClinVar (Landrum et al., 2020). We apply Ensembl VEP (McLaren et al., 2016) to categorize variants by genomic regions into consequence types. Detailed processing is laid out in A.1.6 and A.1.7. Each variant is a genomic position with a mutation $x \in \{A, C, G, T\}$ and a label $y \in \{0, 1\}$. The adjacent 512 bp serve as embedding context.
**Metric**     We compute the AUROC. Additionally, we report separate AUROCs for the variant consequence types to gain further insight into what genomic features are driving performance.

## 4 MODELING

**Language Models**     We benchmark available LMs suitable for the human genome (Table 2). Checkpoint selection criteria are laid out in A.6.2. Additionally, we train two simple baseline DNA LMs: An AWD-LSTM (Merity et al., 2017) model trained on three species, and a dilated CNN similar to GPN (Benegas et al., 2023), trained on the human genome. The model differs from GPN in the parameter count and the length of training sequences (A.6.1).

**Downstream model**     We train a lightweight supervised two-layer CNN model with 64 channels on top of the LM embeddings for each task. LM weights are kept frozen and are not fine-tuned. For LMs with reduced output length due to tokenization, embeddings are upsampled to the original sequence length (A.6.3). For sequence-level tasks, we apply average pooling after the last CNN layer. For the enhancer annotation task, the number of channels was reduced to prevent overfitting. No downstream model is trained for variant effect prediction, as the cosine distance of the LM embeddings directly serves as the zero-shot predictor.

**Supervised baselines**     For each task, we train two supervised models without pre-training. For a direct comparison of raw and embedded DNA, we train the two-layer CNN on one-hot encoded sequences. For chromatin accessibility, histone modificaton and CpG methylation prediction, we train the Basset model (Kelley et al., 2016), which was specifically designed for modeling genome-wide functional genomics data. For gene finding and enhancer annotation, we train the ResNet CNN model on one-hot encoded sequences. For variant effect prediction, no supervised models are trained. For all tasks where Basset is not applicable, we report the performance of a previously published task-specific expert method on the benchmark dataset to put LM performance into context.

**Table 3:** Results on all tasks. The best performing DNA LM for each task is highlighted in bold.

| | | Gene finding | Enhancer annotation | Chromatin accessibility | Histone modification | CpG Methylation | Variant effects (expression) | Variant effects (disease) |
|---|---|---|---|---|---|---|---|---|
| **Expert method** | | 0.80 AUGUSTUS | 0.07 ENFORMER | 0.85 BASSET | 0.74 BASSET | 0.93 BASSET | 0.70 DEEPSEA | 0.56 DEEPSEA |
| **Fully supervised** | **ResNet** | 0.46 | 0.06 | - | - | - | - | - |
| | **CNN** | 0.00 | 0.03 | 0.75 | 0.76 | 0.84 | - | - |
| **Pre-trained** | **ResNet-LM** | 0.36 | 0.02 | 0.82 | 0.77 | 0.87 | 0.55 | 0.55 |
| | **AWD-LSTM** | 0.05 | 0.03 | 0.69 | 0.74 | 0.81 | 0.53 | 0.45 |
| | **NT-H** | 0.41 | 0.05 | 0.74 | 0.76 | 0.88 | 0.55 | 0.48 |
| | **NT-MS** | **0.68** | **0.06** | 0.79 | 0.78 | **0.92** | 0.54 | **0.77** |
| | **NT-1000G** | 0.49 | 0.04 | 0.77 | 0.77 | 0.89 | 0.45 | 0.49 |
| | **NT-V2** | 0.64 | 0.05 | 0.80 | 0.76 | 0.91 | 0.48 | 0.48 |
| | **DNABERT** | 0.20 | 0.03 | **0.85** | **0.79** | 0.91 | **0.60** | 0.56 |
| | **DNABERT-2** | 0.43 | 0.03 | 0.81 | 0.78 | 0.90 | 0.49 | 0.51 |
| | **GENA-LM BERT** | 0.52 | 0.03 | 0.76 | 0.78 | 0.91 | 0.49 | 0.55 |
| | **GENA-LM BigBird** | 0.39 | 0.04 | 0.82 | 0.78 | 0.91 | 0.49 | 0.52 |
| | **HyenaDNA large** | 0.35 | 0.03 | 0.84 | 0.76 | 0.91 | 0.51 | 0.45 |
| | **HyenaDNA tiny** | 0.10 | 0.02 | 0.78 | 0.76 | 0.86 | 0.47 | 0.44 |
| | **GROVER** | 0.28 | 0.03 | 0.82 | 0.77 | 0.89 | 0.56 | 0.51 |

# 5 RESULTS

**Gene finding**   DNA LMs show promising performance for gene finding (Table 3). The two-layer CNN baseline fails to learn, possibly due to its inherent limitation to local context. However, the same CNN is able to achieve varying levels of performance when using LM embeddings, suggesting that embeddings capture some long-range information. NT-MS and NT-V2 outperform all other models by a wide margin, but still do not approach the highly specialized AUGUSTUS (Stanke & Waack, 2003) gene finding model. This highlights that while more specialized downstream models are still needed to accurately predict gene structure, using pre-trained DNA LM embeddings presents a promising avenue to attain good performance. Computing individual performance metrics across all classes (Table A8) reveals that although there is high variance in the performance across all classes, some embeddings capture splice sites fairly considering their low frequency. HyenaDNA-large, although being the only LM whose context length fully covers the input length of the task, only shows modest performance.

**Enhancer annotation**   All investigated models perform poorly on this task. Enhancer annotation is an extremely difficult task due to the length scale, sparsity of the signal, and small dataset, which pose challenges for all investigated models. Although the supervised baseline has a large enough receptive field to detect the long-range interaction, the size of the dataset is prohibitive for performance. The performance of Enformer (Avsec et al., 2021) (A.7) is comparable on this task, but it must be noted that this is an unsupervised method that was not trained directly on enhancer data. Rather, it infers their locations from learning to predict other genome annotations. Predicting gene-specific enhancers from sequence alone without considering supporting experimental data as input therefore remains a highly difficult problem. While this task already proves to be highly challenging for current models at the given length scales, we note that biology is even more complex, with enhancers potentially being millions of bp away.

**Chromatin accessibility**   DNABERT shows the highest performance, on par with the specialized Basset model (0.85). All other LMs perform worse on this task, offering no advantage over Basset.

**Histone modification**   NT-MS and DNABERT show the highest performance (0.74), outperforming Basset (0.72). This suggests that LM embeddings can improve performance for histone modification prediction, albeit at marginal levels.

**CpG methylation**   NT-MS performs best (0.92) on all included cell lines (Table A11), but is outperformed by Basset (0.93). DNABERT, GENA-LM and HyenaDNA-large also perform competitively, indicating that embeddings capture information about CpG island methylation patterns.

**Variant effect prediction**   DNA LMs show some signal for unsupervised prediction of noncoding variant effects. As the two datasets focus on different genomic regions, we only see limited consistency between the expression and disease variant tasks, with DNABERT and NT-MS performing best respectively. While being worse than the supervised DeepSEA method, DNABERT matches DeepSEA's unsupervised performance on the expression dataset (AUROC 0.6, A.7). On the disease

dataset, multiple LMs approach DeepSEA's Disease Impact Score, with NT-MS outperforming it. When dissecting performance by variant types, we find that the performance of NT-MS is driven by variants affecting splice sites and introns (Table A13). While splice sites can be considered noncoding DNA, they are not the focus of DeepSEA, which models chromatin features, and shows stronger performance in UTRs and up- or downstream regions. Similar to the results on the expression dataset, we find that DNABERT outperforms NT-MS in such regions, suggesting that the two LMs learned distinct sequence features during pre-training. As all other NT models show weaker performance on variants affecting gene structure, this could be a consequence of the model's large size and multi-species pre-training. However, we do not see similarly strong performance in the multi-species DNABERT-2 and NT-V2.

## 6 DISCUSSION

We find that currently available DNA LMs already show promising performance on some tasks over fully supervised baselines, but do not offer consistent improvements over all included tasks and can fall short of surpassing specialized existing prediction methods. Overall, we find that NT-MS is a strong default LM, but is in some tasks inferior to the much smaller DNABERT. Interestingly, while both models trained using the MLM objective, we find that they learned distinct genomic features during pre-training. With the pre-training data and the tokenization strategy being the key architectural difference, these choices may deserve more attention in future DNA LMs.

For modeling functional genomics data, DNA LMs only show limited utility. In direct comparison to the Basset model trained on the same data, LM embeddings fail to yield consistent improvements in performance when only using a two-layer CNN.

On the gene finding task, we observe that NT-MS with a simple two-layer CNN shows promising performance compared to the specialized AUGUSTUS, which was found to be the state of the art in a recent benchmark (Scalzitti et al., 2020). This suggests that future more sophisticated LM-based gene finders might become a method of choice for this problem. The result also indicates that current DNA LMs are capable of modeling long-range dependencies to some extent.

Probing LMs at even longer ranges in the enhancer annotation task reveals that long-range understanding still needs improvement for sparse problems with limited data. This highlights a key issue facing DNA LMs: Not only is there a need for long-range modeling to improve our understanding of the genome, as demonstrated by Avsec et al. (2021), but it also raises a fundamental question as to whether current LM training objectives will lead to the incorporation of such distant, sparse signals, or whether the local sequence context is all that is required for sequence reconstruction and some level of supervision is needed. Since BEND is not inherently tied to an LM objective, our standardized benchmark may also prove useful for evaluating eventual DNA representation models that follow a different paradigm.

## 7 LIMITATIONS AND OUTLOOK

As the curation of a comprehensive benchmark task collection requires experimental ground-truth data to be available, and most published models are trained on human data, we focused BEND on the human genome. BEND aims at comparing the effectiveness of different model architectures and training strategies for learning representations from genomic data, under the assumption that other, similarly structured genomes should behave comparably under self-supervision. However, an important question that remains unanswered is whether DNA LMs can aid with generalization across different organisms. In the future, we hope to extend the benchmark to other, diverse organisms, so that generalization power can be tested in a transfer-learning setting, i.e. by training a task on a given organism, and evaluating performance on another.

In BEND, we benchmarked to what extent embeddings capture features that can be leveraged by downstream models for prediction. This approach is fully agnostic regarding the underlying LM's methodology and scales to models of any size. Other works proposed to fine-tune LMs on tasks directly. While this potentially conflates a representation's content with the inductive bias of a model architecture for a given task, fine-tuning may yield performance gains beyond the results observed in this work (Nguyen et al., 2023; Zhou et al., 2023). Another aspect to be investigated in the future is to dive deeper into how LMs learn features during pre-training, as done previously for protein LMs (Vig et al., 2021).

## 8 ACKNOWLEDGEMENTS AND DISCLOSURE OF FUNDING

This work was funded in part by Innovation Fund Denmark (0153-00197B), the Novo Nordisk Foundation through the MLLS Center (Basic Machine Learning Research in Life Science, NNF20OC0062606), the Pioneer Centre for AI (DRNF grant number P1), and the Danish Data Science Academy, which is funded by the Novo Nordisk Foundation (NNF21SA0069429) and VIL-LUM FONDEN (40516).

This work was supported by the Helmholtz Association under the joint research school "Munich School for Data Science (MUDS)".

We would like to acknowledge and thank Ziga Avsec, David Kelley, as well as the rest of the authors behind the Enformer model Avsec et al. (2021), for providing the set of transcription start sites used in the enhancer annotation task.

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

## A  APPENDIX

### A.1  DATASET DOCUMENTATION

We document datasets that were established in this work following the Datasheets for Datasets framework (Gebru et al., 2018), discussing *Motivation*, *Composition*, *Collection process*, *Preprocessing* and *Uses*, as appropriate. As no new experimental data was acquired within this study, and discussing the original experimental protocols would exceed the scope of a datasheet, we limit the *Collection* sections to listing all relevant sources, where experimental procedures are documented. We omit *Distribution* and *Maintenance* as these are identical for each dataset.

#### A.1.1  GENE FINDING

- **Motivation** The dataset was created to benchmark the performance of models on the gene finding task. Given a DNA sequence, a model predicts the structure of the gene, classifying nucleotides as introns, exons, splice sites and noncoding regions.

- **Composition** Instances are the coordinates of human genes including flanking context, together with nucleotide-level labels. There are 9 different labels $y \in \{E_F, D_F, I_F, A_F, E_R, D_R, I_R, A_R, NC\}$ denoting exons, donor splice sites, introns, acceptor splice sites and noncoding nucleotides. $F$ and $R$ denote whether the gene lies on the forward or reverse strand. There are a total of 5,976 instances with instance lengths ranging from 1,433 to 14,000 nucleotides (Figure A1). The dataset is a sample of instances, selected based on the transcript support level of the genes. Label sequences are complete without missing data. A recommended data split is included. The dataset depends on the human reference genome GRCh38.

- **Collection** All data was acquired from GENCODE release 44 (Frankish et al., 2021).

- **Preprocessing** Label sequences were generated from `gff` files downloaded from GENCODE (Frankish et al., 2021). Only HAVANA protein coding gene annotations that were tagged with a transcript support level 1 or 2 from GENCODE as well as level 1 or 2 confidence, meaning that the transcript is experimentally verified, were considered. For genes with alternative splicing, only the transcript with the best level of experimental support was chosen. In cases where support was equal, a random transcript was chosen. For each transcript, flanking context to include was sampled at random. Following AUGUSTUS' recommendations[1] for training and testing gene finding models, the data was split so that no pair of instances in different partitions shares more than 80% sequence identity of the mature protein. GraphPart (Teufel et al., 2023) with Needleman-Wunsch global sequence alignments was used for splitting at a 80% sequence identity into train (80% of the data), test and validation (10% each).

- **Uses** The specific dataset was established in this study and not used before. Data from GENCODE has seen widespread use.

#### A.1.2  ENHANCER ANNOTATION

- **Motivation** This dataset was created to benchmark the performance of models in annotating the correct enhancer segment. Given a DNA sequence starting at the transcription start site of a gene and encompassing the enhancer, each nucleotide is classified based on a binary task into enhancer or non-enhancer.

- **Composition** Instances are coordinates in the human genome, covering 100,096 nucleotides each, associated with binary label sequences of length 782. Instances are centered on the transcription start site of a gene and extend in both directions symmetrically, containing the enhancer element on one side (Figure A2). In the label sequence, each label applies to a binned segment of 128 bp. The segment is labeled 1 if it contains a nucleotide lying in the enhancer element, and 0 otherwise (Figure A3). Some genes have multiple enhancer elements. In these cases all enhancer elements are labelled in one sample.

- **Collection** Enhancer locations for genes of interest are obtained from the CRISPR interference (CRISPRi) experiments of Fulco et al. (2019) and Gasperini et al. (2019) (GEO accession GSE120861) via Avsec et al. (2021). CRISPRi experiments perturb a candidate enhancer

---

[1]`https://vcru.wisc.edu/simonlab/bioinformatics/programs/augustus/docs/`
`tutorial2015/training.html`

and record whether the perturbation resulted in a change in gene expression. These experiments thereby directly measure the connection of an enhancer element to a specific gene. Following Avsec et al., we consider enhancers that had an expression change as "validated". Enhancer-gene pairs that were predicted by the activity-by-contact (ABC) method only were not considered experimentally validated and excluded. For each gene, the predicted main transcription start site was obtained directly from Avsec et al. (2021).

- **Preprocessing** All non-validated gene-enhancer pairs were discarded, as were all pairs with over 50,048 bp between the enhancer element and the transcription start site. Samples were split chromosome-wise into 10 partitions for cross-validation (1: chr7, chr8, chr18; 2: chr10, chrX, chr13; 3: chr14, chr22, chr6; 4: chr20, chr3; 5: chr11, chr12; 6: chr19; 7: chr4, chr5; 8: chr15, chr21, chr2; 9: chr1, chr16; 10: chr17, chr9).

- **Uses** The binned label sequences over 100,096 bp were established in this work. The same underlying enhancer-gene pairs were amongst the ones used in Avsec et al. (2021).

### A.1.3 CHROMATIN ACCESSIBILITY PREDICTION

- **Motivation** The data was created to benchmark the performance of models on the chromatin accessibility prediction task. Given a DNA sequence, a model predicts whether the sequence is in open or closed chromatin in different cell types.

- **Composition** Instances are coordinates in the human genome, covering 512 nucleotides each, associated with a binary label vector $\mathbf{y} \in \{0, 1\}^{125}$, indicating whether the DNA is in open (1) or closed (0) chromatin in 125 cell types (Table A2). This state is determined experimentally by whether the window of 512 nucleotides contains a DNAse I hypersensitive site. There are 2,005,617 instances.

- **Collection** Data was obtained from ENCODE (ENCODE Project Consortium, 2012; Luo et al., 2020; Kagda et al., 2023; Hitz et al., 2023). We downloaded DNase I hypersensitivity peaks for 125 cell types in `bed` format.

- **Preprocessing** The preprocessing followed Kelley et al. (Kelley et al., 2016). Peaks were extended from to 512 bp from their midpoint, and peaks overlapping by less than 200bp were merged greedily. When peaks of two or more cell types were merged, the resulting sample was annotated with multiple cell type labels. Samples were split chromosome-wise into test (chr1, chr8, chr9; 372,153 samples), validation (chr2, chr4; 279,422 samples) and train (remaining chromsomes; 1,354,042 samples).

- **Uses** The specific dataset was established in this study and not used before. Data from ENCODE has seen widespread use, and comparable datasets were originally created in (Kelley et al., 2016).

### A.1.4 HISTONE MODIFICATION PREDICTION

- **Motivation** This dataset benchmarks the ability of models to predict post-translational modifications of Histone proteins. Given a DNA sequence, a model is tasked to predict which histone-modifications are present in the underlying nucleosome.

- **Composition** Instances are coordinates in the human genome, covering 512 nucleotides each, associated with a binary label vector of size 18, indicating whether a given histone mark (Table A1) is present (1) or not (0).

- **Collection** Data was obtained from ENCODE (ENCODE Project Consortium, 2012). Narrow peaks files of 18 Histone ChIP-seq experiments was gathered from ENCODE (ENCODE Project Consortium, 2012) in `bed` format.

- **Preprocessing** Following Kelley et al. (2016), peaks were extended from to 512 bp from their midpoint, with peaks overlapping by less than 200bp being merged greedily. When peaks of two or more ChIP-seq experiments were merged, the resulting sample was annotated with the label of each experiment. Note that some Histone marks were covered by multiple experiments. Samples were split chromosome-wise into test (chr1, chr8, chr9; 120,567 samples), validation (chr2, chr4; 70,801 samples) and train (remaining chromsomes; 420,713 samples).

- **Uses** The specific dataset was established in this study and not used before. It is based on publicly available Histone ChIP-seq dataset from the ENCODE project, has seen widespread use.

A.1.5  CPG METHYLATION

- **Motivation** This dataset benchmarks the ability of models to predict the methylation of CpG sites. Methylation is an epigenetic modification of DNA that can affect a sequence's activity and repress gene expression. The methylation of a C to form 5-methylcytosine in CpG sites is the most prominent type of methylation.

- **Composition** Instances are coordinates in the human genome, covering 512 nucleotides each, associated with a binary label vector of size 19, indicating whether the CpG site at the center of the segment is methylated (1) or not (0) in a given cell line (Table A3).

- **Collection** We gathered 7 human cell line whole-genome shotgun bisulfite sequencing (WGBS) experiments from ENCODE (ENCODE Project Consortium, 2012) and processed the "methylation state at CpG" `bed` files. To select cell lines, experiments marked in ENCODE as "Extremely low coverage" or "Insufficient coverage" were excluded.

- **Preprocessing** We removed all CpG sites that lie on non-standard chromosomes and that have a variant in the respective sample genome that does not match the reference genome. Following DeepCpG (Angermueller et al., 2017), we removed CpG sites that are covered by less than 4 reads. CpG sites that had at least 90% methylated reads were labeled as methylated, sites with less than 10% methylated reads were labeled as unmethylated, remaining sites were discarded. We took the common subset of CpG sites passing the filtering criteria in all 7 experiments. Sites that were not measured in all experiments were discarded, obtaining 959,039 sites in total. CpG sites were extended with flanking context to yield 512bp windows centered on the CpG site. Samples were split by chromosomes (test: chr4, chr13, chr19, chr21 - 106,227 samples; validation: chr5, chr9, chr22 - 109,717 samples; remainder train - 743,095 samples).

- **Uses** The specific dataset was established in this study and not used before. It is based on publicly available WGBS data from the ENCODE project which has seen widespread use.

A.1.6  NONCODING VARIANT EFFECTS (EXPRESSION)

- **Motivation** The dataset was created to benchmark the zero-shot noncoding variant effect prediction performance of models. Given a reference nucleotide, and a mutated nucleotide, two embeddings are computed and their cosine distance is used as the predictor.

- **Composition** Instances are single nucleotide polymorphisms (SNP), genetic coordinates with a reference nucleotide $x_{ref} \in \{A, C. G, T\}$ and a variant $x_{var} \in \{A, C. G, T\}$ together with a binary label $y \in \{0, 1\}$ indicating whether the SNP has an effect on gene expression (1) or is genetic background variation (0). We use the same SNPs included in DeepSEA (Zhou & Troyanskaya, 2015). The discovery of such functional SNPs, so-called eQTLs (Expression quantitative trait loci) is done through large-scale genetics studies that link genetic variation to gene expression. There are 98,221 background SNPs and 8,000 variants with effect in total. As this is a zero-shot task, no split is required. The dataset depends on the human reference genome GRCh38. eQTLs were collected from GRASP (Leslie et al., 2014) and background SNPs from the 1000 Genomes Project (McVean et al., 2012). The dataset is a subsample of SNPs present in these databases. While the 1000 Genomes Project aims at faithfully representing human genetic variation, it might still suffer from ethnicity biases (Table A6). The GRASP database is biased towards eQTLs observed in individuals with european ancestry (Table A7).

- **Collection** Genomic coordinates for SNPs were taken from DeepSEA (Zhou & Troyanskaya, 2015).

- **Preprocessing** As the original genomic coordinates refer to the previous reference genome GRCh37, we used LiftOver to transfer the coordinates to the current reference GRCh38. Any coordinates that could not be mapped were discarded. Variants where the original reference nucleotide does not match the nucleotide at the indicated position in GRCh38 were removed. We only use SNPs included in fold 0. We applied Ensembl VEP (McLaren et al., 2016) to categorize variants by consequence. VEP infers the consequence of a variant by comparing a variant's position to the reference genome annotation, determining what type of sequence region (Table A4) the variant lies in. To obtain one consequence per variant, we use VEP's `--most_severe` flag, returning the consequence with the potentially most severe effect on function. In DeepSEA, the adjacent 1,000 bp served as context for classification. As this exceeds the maximum context length

of some of the benchmarked models, and chunking inputs is not a meaningful strategy when adjacent bps serve only as context for an unsupervised embedding, we use 512 bp instead. As this is a zero-shot task, no split is performed, with the full dataset serving as test set.

- **Uses** The same SNPs on GRCh37 were originally used in DeepSEA for both unsupervised (zero-shot) and supervised variant effect prediction.

### A.1.7 Noncoding variant effects (Disease)

- **Motivation** The dataset was created to benchmark the zero-shot noncoding variant effect prediction performance of models. Given a reference nucleotide, and a mutated nucleotide, two embeddings are computed and their distance is used as the predictor.

- **Composition** Instances are single nucleotide polymorphisms (SNP), genetic coordinates with a reference nucleotide $x_{ref} \in \{A, C. G, T\}$ and a variant $x_{var} \in \{A, C. G, T\}$ together with a binary label $y \in \{0, 1\}$ indicating whether the SNP is benign (0) or pathogenic (1). There are 274,399 benign and 21,524 pathogenic SNPs in total. As this is a zero-shot task, no split is required. The dataset depends on the human reference genome GRCh38.

- **Collection** SNPs annotated as (likely) benign or pathogenic were collected from ClinVar, using the `variant_summary` file from 2023-07-02 (Landrum et al., 2020). We collected all variants annotated as `single nucleotide variant` with a review status of at least one star.

- **Preprocessing** To subset ClinVar for noncoding variants, we first discarded all variants that are annotated as being in a protein in ClinVar itself. To further remove variants whose molecular effect might not be annotated in ClinVar, we compared each SNP to GENCODE release 43 (Frankish et al., 2021). All SNPs that were found to be in a CDS, start codon or stop codon were considered coding and removed. We omit SNPs in the mitochondrial genome ("chromosome M") as they are incompatible with the DeepSEA literature baseline. Following Frazer et al. (2021), the annotations "Likely pathogenic", "Pathogenic" and "Likely benign", "Benign" were combined to yield binary labels. We applied Ensembl VEP (McLaren et al., 2016) to categorize variants by consequence. VEP infers the consequence of a variant by comparing a variant's position to the reference genome annotation, determining what type of sequence region (Table A5) the variant lies in. To obtain one consequence per variant, we use VEP's `--most_severe` flag, returning the consequence with the potentially most severe effect on function. The adjacent 512 bp serve as context for embedding. As this is a zero-shot task, no split is performed, with the full dataset serving as test set.

- **Uses** The specific dataset was established in this study and not used before. Data from ClinVar has seen widespread use for variant effect prediction.

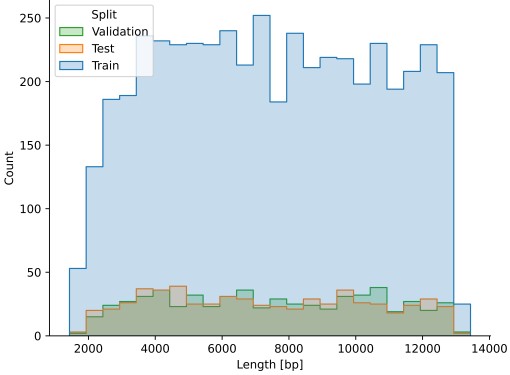

**Figure A1:** Length distribution of samples in the gene finding dataset.

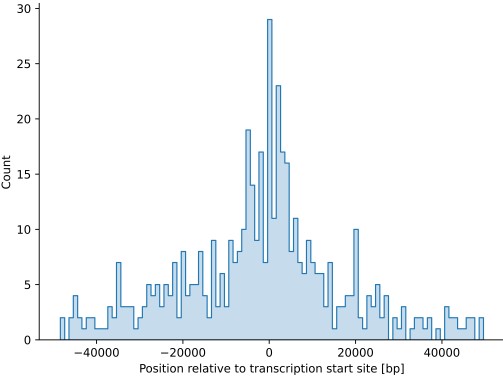

**Figure A2:** Distance to main TSS distribution of the enhancer elements in the enhancer annotation dataset.

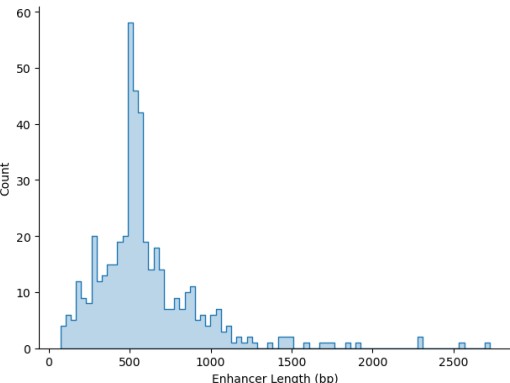

**Figure A3:** Length distribution of the enhancer elements in the dataset.

**Table A1:** Detailed label composition of the histone modification multilabel dataset (n=625,229).

| ENCODE modification | Label ID | # positive instances | % positive |
|---|---|---|---|
| H3K27me3_K562 | 0 | 41,506 | 6.64% |
| H3K9ac_K562 | 1 | 93,261 | 14.92% |
| H3K9me3_K562 | 2 | 25,295 | 4.05% |
| H3K4me1_K562 | 3 | 98,678 | 15.78% |
| H3K9ac_K562 | 4 | 35,382 | 5.66% |
| H3K4me1_K562 | 5 | 92,587 | 14.81% |
| H3K36me3_K562 | 6 | 71,400 | 11.42% |
| H3K36me3_K562 | 7 | 69,975 | 11.19% |
| H4K20me1_K562 | 8 | 38,312 | 6.13% |
| H3K27me3_K562 | 9 | 133,535 | 21.36% |
| H3K4me3_K562 | 10 | 21,717 | 3.47% |
| H3K4me3_K562 | 11 | 19,706 | 3.15% |
| H3K4me3_K562 | 12 | 29,394 | 4.70% |
| H3K4me3_K562 | 13 | 40,934 | 6.55% |
| H3K79me2_K562 | 14 | 67,714 | 10.83% |
| H3K4me2_K562 | 15 | 59,069 | 9.45% |
| H3K27ac_K562 | 16 | 42,993 | 6.88% |
| H2AFZ_K562 | 17 | 107,810 | 17.24% |

**Table A2:** Detailed label composition of the chromatin accessibility multilabel dataset (n=2,062,128).

| ENCODE cell line | Label ID | # positive instances | % positive |
|---|---|---|---|
| 8988T | 0 | 184,985 | 8.97% |

| ENCODE cell line | Label ID | # positive instances | % positive |
| --- | --- | --- | --- |
| AoSMC | 1 | 158,918 | 7.71% |
| Chorion | 2 | 171,737 | 8.33% |
| CLL | 3 | 89,723 | 4.35% |
| Fibrobl | 4 | 394,288 | 19.12% |
| FibroP | 5 | 249,221 | 12.09% |
| Gliobla | 6 | 158,628 | 7.69% |
| GM12891 | 7 | 135,186 | 6.56% |
| GM12892 | 8 | 149,741 | 7.26% |
| GM18507 | 9 | 109,689 | 5.32% |
| GM19238 | 10 | 142,111 | 6.89% |
| GM19239 | 11 | 120,883 | 5.86% |
| GM19240 | 12 | 174,077 | 8.44% |
| H9ES | 13 | 154,898 | 7.51% |
| HeLa-S3_IFNa4h | 14 | 109,698 | 5.32% |
| Hepatocytes | 15 | 164,799 | 7.99% |
| HPDE6-E6E7 | 16 | 132,643 | 6.43% |
| HSMM_emb | 17 | 123,566 | 5.99% |
| HTR8svn | 18 | 122,358 | 5.93% |
| Huh-7.5 | 19 | 172,276 | 8.35% |
| Huh-7 | 20 | 142,675 | 6.92% |
| iPS | 21 | 192,872 | 9.35% |
| Ishikawa_Estradiol | 22 | 131,324 | 6.37% |
| Ishikawa_4OHTAM | 23 | 133,612 | 6.48% |
| LNCaP_androgen | 24 | 138,434 | 6.71% |
| MCF-7_Hypoxia | 25 | 146,053 | 7.08% |
| Medullo | 26 | 218,010 | 10.57% |
| Melano | 27 | 276,645 | 13.42% |
| Myometr | 28 | 165,059 | 8.00% |
| Osteobl | 29 | 367,127 | 17.80% |
| PanIsletD | 30 | 198,709 | 9.64% |
| PanIslets | 31 | 172,141 | 8.35% |
| pHTE | 32 | 262,572 | 12.73% |
| ProgFib | 33 | 201,038 | 9.75% |
| RWPE1 | 34 | 146,568 | 7.11% |
| Stellate | 35 | 157,369 | 7.63% |
| T-47D | 36 | 140,932 | 6.83% |
| CD4_Th0 | 37 | 195,611 | 9.49% |
| Urothelia | 38 | 136,076 | 6.60% |
| Urothelia_UT189 | 39 | 169,356 | 8.21% |
| AG04449 | 40 | 163,835 | 7.94% |
| AG04450 | 41 | 145,390 | 7.05% |
| AG09309 | 42 | 198,670 | 9.63% |
| AG09319 | 43 | 139,005 | 6.74% |
| AG10803 | 44 | 168,529 | 8.17% |
| AoAF | 45 | 171,356 | 8.31% |
| BE2_C | 46 | 172,185 | 8.35% |
| BJ | 47 | 160,706 | 7.79% |
| Caco-2 | 48 | 118,338 | 5.74% |
| CD20+ | 49 | 100,298 | 4.86% |
| CD34+ | 50 | 158,606 | 7.69% |
| CMK | 51 | 129,859 | 6.30% |
| GM06990 | 52 | 88,680 | 4.30% |
| GM12864 | 53 | 132,999 | 6.45% |
| GM12865 | 54 | 139,644 | 6.77% |
| H7-hESC | 55 | 263,281 | 12.77% |
| HAc | 56 | 177,288 | 8.60% |
| HAEpiC | 57 | 201,958 | 9.79% |

| ENCODE cell line | Label ID | # positive instances | % positive |
| --- | --- | --- | --- |
| HA-h | 58 | 197,746 | 9.59% |
| HA-sp | 59 | 188,882 | 9.16% |
| HBMEC | 60 | 197,261 | 9.57% |
| HCF | 61 | 171,925 | 8.34% |
| HCFaa | 62 | 182,168 | 8.83% |
| HCM | 63 | 190,478 | 9.24% |
| HConF | 64 | 150,615 | 7.30% |
| HCPEpiC | 65 | 207,114 | 10.04% |
| HCT-116 | 66 | 110,464 | 5.36% |
| HEEpiC | 67 | 206,638 | 10.02% |
| HFF | 68 | 189,177 | 9.17% |
| HFF-Myc | 69 | 206,882 | 10.03% |
| HGF | 70 | 143,241 | 6.95% |
| HIPEpiC | 71 | 222,312 | 10.78% |
| HL-60 | 72 | 158,336 | 7.68% |
| HMF | 73 | 176,498 | 8.56% |
| HMVEC-dAd | 74 | 120,737 | 5.85% |
| HMVEC-dBl-Ad | 75 | 159,641 | 7.74% |
| HMVEC-dBl-Neo | 76 | 164,741 | 7.99% |
| HMVEC-dLy-Ad | 77 | 124,355 | 6.03% |
| HMVEC-dLy-Neo | 78 | 149,601 | 7.25% |
| HMVEC-dNeo | 79 | 137,163 | 6.65% |
| HMVEC-LBl | 80 | 167,109 | 8.10% |
| HMVEC-LLy | 81 | 141,044 | 6.84% |
| HNPCEpiC | 82 | 209,477 | 10.16% |
| HPAEC | 83 | 119,805 | 5.81% |
| HPAF | 84 | 185,109 | 8.98% |
| HPdLF | 85 | 168,839 | 8.19% |
| HPF | 86 | 151,615 | 7.35% |
| HRCEpiC | 87 | 189,381 | 9.18% |
| HRE | 88 | 184,386 | 8.94% |
| HRGEC | 89 | 134,424 | 6.52% |
| HRPEpiC | 90 | 224,149 | 10.87% |
| HVMF | 91 | 167,746 | 8.13% |
| Jurkat | 92 | 155,987 | 7.56% |
| Monocytes-CD14+ | 93 | 131,745 | 6.39% |
| NB4 | 94 | 140,287 | 6.80% |
| NH-A | 95 | 188,983 | 9.16% |
| NHDF-Ad | 96 | 227,566 | 11.04% |
| NHDF-neo | 97 | 185,464 | 8.99% |
| NHLF | 98 | 203,663 | 9.88% |
| NT2-D1 | 99 | 179,350 | 8.70% |
| PANC-1 | 100 | 114,230 | 5.54% |
| PrEC | 101 | 164,299 | 7.97% |
| RPTEC | 102 | 166,607 | 8.08% |
| SAEC | 103 | 195,586 | 9.48% |
| SKMC | 104 | 203,116 | 9.85% |
| SK-N-MC | 105 | 142,957 | 6.93% |
| SK-N-SH_RA | 106 | 86,739 | 4.21% |
| Th2 | 107 | 86,210 | 4.18% |
| WERI-Rb-1 | 108 | 188,325 | 9.13% |
| WI-38 | 109 | 163,827 | 7.94% |
| WI-38_4OHTAM | 110 | 202,173 | 9.80% |
| A549 | 111 | 161,511 | 7.83% |
| GM12878 | 112 | 168,725 | 8.18% |
| H1-hESC | 113 | 241,281 | 11.70% |
| HeLa-S3 | 114 | 183,717 | 8.91% |

| ENCODE cell line | Label ID | # positive instances | % positive |
| --- | --- | --- | --- |
| HepG2 | 115 | 180,213 | 8.74% |
| HMEC | 116 | 321,049 | 15.57% |
| HSMM | 117 | 291,793 | 14.15% |
| HSMMtube | 118 | 304,753 | 14.78% |
| HUVEC | 119 | 179,245 | 8.69% |
| K562 | 120 | 190,083 | 9.22% |
| LNCaP | 121 | 291,954 | 14.16% |
| MCF-7 | 122 | 188,759 | 9.15% |
| NHEK | 123 | 201,376 | 9.77% |
| Th1 | 124 | 293,092 | 14.21% |

**Table A3:** Detailed label composition of the CpG methylation multilabel dataset.

| ENCODE cell line | Label ID | % methylated |
|---|---|---|
| SK-N-SH | 0 | 83% |
| GM23248 | 1 | 84% |
| A549 | 2 | 83% |
| HepG2 | 3 | 81% |
| HUES64 | 4 | 91% |
| GM23248 | 5 | 84% |
| HeLa-S3 | 6 | 84% |

**Table A4:** VEP variant consequence categories in the expression variant effects dataset.

| Consequence | Background | eQTL | % eQTL |
|---|---|---|---|
| **Intron variant** | 55,710 | 5,002 | 8.24% |
| **Intergenic variant** | 22,465 | 753 | 3.24% |
| **Upstream gene variant** | 5,760 | 579 | 9.13% |
| **Downstream gene variant** | 4,146 | 435 | 9.50% |
| **Regulatory region variant** | 3,762 | 248 | 6.18% |
| **Noncoding transcript exon variant** | 2,757 | 342 | 11.03% |
| **3' UTR variant** | 1,599 | 426 | 21.03% |
| **5' UTR variant** | 408 | 54 | 11.69% |
| **TF binding site variant** | 410 | 23 | 5.31% |
| **Splice region variant** | 99 | 30 | 23.26% |
| **splice polypyrimidine tract variant** | 85 | 18 | 17.48% |
| **Missense variant** | 51 | 9 | 15.00% |
| **Splice donor region variant** | 27 | 4 | 12.90% |
| **Synonymous variant** | 20 | 4 | 16.67% |
| **Splice donor variant** | 13 | 2 | 13.33% |
| **Splice donor 5th base variant** | 6 | 7 | 53.85% |
| **Splice acceptor variant** | 5 | 0 | 0.00% |
| **mature miRNA variant** | 2 | 0 | 0.00% |
| **Stop lost variant** | 1 | 1 | 50.00% |

**Table A5:** VEP variant consequence categories in the disease variant effects dataset.

| Consequence | Benign | Pathogenic | % Pathogenic |
|---|---|---|---|
| **Intron variant** | 138,023 | 188 | 0.14% |
| **Splice region variant** | 40,040 | 320 | 0.79% |
| **splice polypyrimidine tract variant** | 39,501 | 185 | 0.47% |
| **Noncoding transcript exon variant** | 23,651 | 70 | 0.30% |
| **3' UTR variant** | 20,407 | 34 | 0.17% |
| **5' UTR variant** | 6,933 | 63 | 0.90% |
| **Upstream gene variant** | 2,245 | 19 | 0.84% |
| **Splice donor region variant** | 1,744 | 312 | 15.18% |
| **Splice donor 5th base variant** | 507 | 553 | 52.17% |
| **Downstream gene variant** | 268 | 4 | 1.47% |
| **-** | 262 | 0 | 0.00% |
| **Splice acceptor variant** | 194 | 9,086 | 97.91% |
| **Splice donor variant** | 189 | 10,622 | 98.25% |
| **mature miRNA variant** | 39 | 1 | 2.50% |
| **Intergenic variant** | 19 | 1 | 5.00% |
| **Regulatory region variant** | 10 | 0 | 0.00% |
| **Synonymous variant** | 3 | 0 | 0.00% |
| **Missense variant** | 1 | 0 | 0.00% |
| **TF binding site variant** | 1 | 0 | 0.00% |

**Table A6:** Population statistics of the 1000 Genomes Project (Phases 1 and 3) The data is based on Supplementary Information Table 1 from Auton et al. (2015).

| Population | Count |
|---|---|
| Gambian in Western Division, The Gambia - Mandinka | 113 |
| Mende in Sierra Leone | 85 |
| Esan in Nigeria | 99 |
| Colombian in Medellin, Colombia | 174 |
| Peruvian in Lima, Peru | 85 |
| Punjabi in Lahore, Pakistan | 96 |
| Iberian populations in Spain | 121 |
| Toscani in Italy | 205 |
| Mexican Ancestry in Los Angeles, California | 130 |
| Sri Lankan Tamil in the UK | 102 |
| Indian Telugu in the UK | 102 |
| British in England and Scotland | 180 |
| Yoruba in Ibadan, Nigeria | 196 |
| Japanese in Tokyo, Japan | 193 |
| Utah residents (CEPH) with Northern and Western European ancestry | 184 |
| Han Chinese in Beijing, China | 200 |
| Chinese Dai in Xishuangbanna, China | 93 |
| Luhya in Webuye, Kenya | 196 |
| Gujarati Indians in Houston, TX | 103 |
| African Ancestry in Southwest US | 122 |
| Finnish in Finland | 192 |
| Han Chinese South | 205 |
| Kinh in Ho Chi Minh City, Vietnam | 99 |
| Bengali in Bangladesh | 86 |
| Puerto Rican in Puerto Rico | 159 |
| African Caribbean in Barbados | 96 |

**Table A7:** Population statistics of the eQTLs in the GRASP 2.0.0.0 database. GRASP combines results from 2,082 individual studies. The ancestry information (`GWASancestryDescription`) is recorded on a study-wide level.

| Ancestry | eQTLs |
|---|---|
| European | 446,403 |
| Mixed | 128,301 |
| Unspecified | 111,218 |
| European/Unspecified | 11,376 |
| African | 2067 |
| Native | 205 |

## A.2 FORMATTING

Building upon established standards in genomics, we curate all tasks in the same format for ease of reuse. Typically, it is not necessary to store DNA sequences $X$ explicitly for each task, as many tasks will refer to the same reference genome. Therefore, for each task, we list the genome coordinates for each sample in a `bed` genome annotation file. Splits and labels $Y$ are also stored in these files, unless they are too complex to be stored in text format and are provided in a `hdf5` file that shares its index with the `bed` file. The `bed`-based format also makes it convenient to include more flanking context of the segments to be predicted without reprocessing the data, should future works find it useful to take more bp into account.

Code to extract DNA sequences from the reference genome with the `bed` coordinates, dataloaders, models and config files is available on Github (https://anonymous.4open.science/r/BEND-8C42/README.md).

## A.3 LICENSE

As far as applicable, our contributions are licensed as *CC BY 4.0*. As no new data was generated in this study, the respective use/redistribution agreements and any copyright claims on the underlying data sources (GENCODE (Frankish et al., 2021), ENCODE (ENCODE Project Consortium, 2012), GRASP (Leslie et al., 2014), 1000 Genomes Project (McVean et al., 2012), Gasperini et al. (2019) (GEO accession GSE120861), Fulco et al. (2019), Avsec et al. (2021)) apply to the provided datasets. Therefore, citation of the original sources is required when using the data provided with BEND. Citations in BibTex format are listed in the BEND repository.

## A.4 DISTRIBUTION

All data is available at `https://sid.erda.dk/cgi-sid/ls.py?share_id=aNQa0Oz2lY` Code, configs and scripts to extract data and run all experiments are provided at `https://github.com/frederikkemarin/BEND`.

## A.5 SOCIETAL IMPACT

Predictors building upon DNA LMs may prove useful in a wide range of biomedical research applications. Moreover, given their promising performance for understanding the effects of variants, future LMs or derived predictors with even higher performance may eventually become relevant for medical applications. If LM-based predictors are used in clinical diagnostics on humans, it is important to ensure that their performance is evaluated over different populations and potential subpopulation biases are accounted for. Moreover, should genomes from human individuals that are not publicly released be used for pre-training LMs, it is important to ensure that their consent is obtained.

## A.6 LM DETAILS

### A.6.1 LMs TRAINED IN THIS WORK

**AWD-LSTM** We trained an autoregressive AWD-LSTM LM using truncated backpropagation through time with a backpropagation window of 100 bps. Starting points were sampled randomly in the genome, and sequences processed until encountering a chromosome end, upon which the hidden state was reset. The model was trained on the full genomes of *H. sapiens*, *M. musculus* and *D. melanogaster* with a batch size of 1,024 for 1 million steps. This represents a minimal multi-species scenario that was selected due to computational constraints. The model has 3 LSTM layers with dimensions 64, 1,024 and 64. Sequences were tokenized on the nucleotide level, yielding an alphabet of size 4. The model was trained on a single NVIDIA RTX 6000 GPU on a local cluster for 35 days.

**Dilated ResNet LM** We trained a dilated CNN with residual connections, which is the same architecture used by GPN (Benegas et al., 2023). Since this model has a large receptive field due to the dilations, we decided to take advantage of this by increasing the length of the training sequences from 512 nucleotides in GPN to 10,000 nucleotides here. To trade off the computational requirements, we reduce the number of hidden channels in the model from 512 to 256. The model was trained by randomly sampling training sequences from contigs of the human reference genome. The reverse complementary of the sampled sequences was used with a 50% chance. Two chromosomes were held out for testing and validation respectively. The model was trained with a batch size of 512 for a total of 50k steps, using 4 NVIDIA A40 GPUs for 14 days.

### A.6.2 LM CHECKPOINT SELECTION

We aimed to cover all DNA LM works that are publicly available and that included the human genome in their pre-training data. For works that introduce more than one pre-trained checkpoint for their proposed LM architectures, we choose a limited number of representative checkpoints in order to make efficient use of available computational resources. Whenever possible, the selection is driven by results and recommendations presented in the original work.

- **DNABERT** We use the checkpoint that tokenizes DNA as overlapping 6-mers. The original DNABERT paper (Ji et al., 2021) states that the 6-mer checkpoint showed the best performance when fine-tuning on the included tasks.

- **Nucleotide Transformer** We evaluate the checkpoints trained on the human reference genome (500M parameters), the 1000 Genomes Project (2.5B parameters) and on the set of genomes from multiple species (2.5B parameters).

- **GENA-LM** In order to include one representative checkpoint both for the BigBird and the BERT architectures, we use `bert-large-t2t` and `bigbird-base-t2t`.

- **HyenaDNA** HyenaDNA provides multiple sizes of the same model architecture trained on the same data. The checkpoints also differ in the length of the sequences they were trained on. We use `tiny-1k`, the smallest checkpoint that was trained on 1,000bp sequences, and `large-1m`, the largest checkpoint trained on 1 million bp sequences.

- **Nucleotide Transformer V2** We evaluate the largest available model with 500M parameters.

### A.6.3 UPSAMPLING OF EMBEDDINGS

LMs that make use of k-mer or byte-pair encoding (BPE) tokenization strategies return less embedding vectors than their original input sequence length. For nucleotide-level prediction tasks, an embedding sequence of equal length to the nucleotide-wise label sequence is needed. In order to benchmark all LMs equally, regardless of how they tokenize inputs, we upsample embeddings. For the 6-mer tokenization employed by NT, we repeat each embedding vector 6 times. For BPE in GENA-LM, DNABERT-2 and GROVER, we repeat each token's embedding by the length of the token's sequence. DNABERT, which uses overlapping k-mers, returns a reduced number of embeddings due to the fact that at the left and right borders of the sequence there is no k/2 context available to construct a k-mer embedding around the nucleotide. As DNABERT does not perform any padding to correct for this, these initial and terminal k-mer embeddings are missing. We repeat the first and the last embedding to match the original input sequence length. For k=6, we repeat the first embedding two and the last embedding three times.

### A.7 TASK DETAILS

Computations for all tasks were performed on single GPUs of the types RTX 6000, RTX 8000, V100, A40 and A100 on local clusters, depending on availability.

**Gene finding** CNN models were trained using AdamW with a learning rate of 0.003 and a weight decay of 0.01 for 100 epochs with a batch size of 64.

AUGUSTUS performance was evaluated on the test set. For each input sequence, exactly one complete gene model was predicted. Since AUGUSTUS only returns the CDS borders as well as the strand, the remaining labels where inferred from the the CDS locations to compare with the ground truth labels. All nucleotides prior to the first CDS and subsequent to the last are labeled as intergenic. Nucleotides between two CDS segments are labeled as introns. The first and last nucleotide of each intron is labeled as a donor and acceptor site respectively for genes predicted to be on the positive strand, on the negative strand it is reversed (acceptor site is the first nucleotide and donor the last).

Augustus was run with the following settings:

```
--strand=both --UTR=off --AUGUSTUS_CONFIG_PATH=path
--gff3=on --genemodel=exactlyone --species=human sequence.fasta
```

**Histone modification** CNN models were trained using AdamW with a learning rate of 0.003 and a weight decay of 0.01 for 100 epochs with a batch size of 256.

**CpG methylation**   CNN models were trained using AdamW with a learning rate of 0.003 and a weight decay of 0.01 for 100 epochs with a batch size of 256.

**Enhancer annotation**   CNN models with channel size 2 were trained using AdamW with a learning rate of 0.001 and a weight decay of 0.01 for 100 epochs with a batch size of 8. Due to the high label imbalance in the data, positive labels were up-weighted in the loss with a weight corresponding to the average fraction of positive to negative labels.

Enformer performance was evaluated using the code provided in the *Compute contribution scores* section of the Enformer notebook[2]. For each sample of 100,086 bp, context was expanded bidirectionally and Enformer contribution scores were obtained. The scores were trimmed back to the original 100,086 bp and average pooled at 128bp, yielding a sequence of 782 bins for each sample.

**Noncoding variant effects**   There are two ways of extracting an embedding for a variant sequence: It is possible to either take the mean embedding of the full context window, or extract the embedding at the position where the SNP is found. Within BEND, we opted for the latter approach, as we consider it more universally applicable to e.g. autoregressive models where preceding embeddings in the context window cannot contain any information on the variant that comes later in the sequence. For NT, this means taking the embedding of the 6-mer token containing the variant. For DNABERT-2 and GENA-LM, the embedding of the BPE token containing the variant is used. For DNABERT with 6-mer tokenization, we use the embedding of the token that has the mutated residue as its 3rd nucleotide.

As in autoregressive models subsequent tokens cannot affect already computed embeddings, we only used an unidirectional context of 512 preceding nucleotides for AWD-LSTM and HyenaDNA.

For the expression dataset, supervised DeepSEA performance was computed from the cross-validated predictions for split 0 available in the supplementary material of the original DeepSEA publication (Zhou & Troyanskaya, 2015). Unsupervised performance could not be recomputed and was taken at 0.6 from DeepSEA's Supplementary Figure 6 for the expression dataset. For the disease dataset, supervised DeepSEA performance was computed by submission to DeepSEA's online version, using the `Beluga` model. The Disease Impact Score (DIS) output was used for benchmarking.

---

[2]`https://github.com/google-deepmind/deepmind-research/blob/master/enformer/enformer-usage.ipynb`

## A.8 EXTENDED RESULTS

**Table A8:** Gene finding recall and precision per label.

| Model | CDS_F (0) Recall | Precision | Donor_F (1) Recall | Precision | Intron_F (2) Recall | Precision | Acceptor_F (3) Recall | Precision | CDS_R (4) Recall | Precision | Acceptor_R (5) Recall | Precision | Intron_R (6) Recall | Precision | Donor_R (7) Recall | Precision | Intergenic (8) Recall | Precision |
|---|---|---|---|---|---|---|---|---|---|---|---|---|---|---|---|---|---|---|
| AUGUSTUS | 0.89 | 0.90 | 0.80 | 0.88 | 0.83 | 0.87 | 0.79 | 0.86 | 0.89 | 0.91 | 0.81 | 0.86 | 0.85 | 0.89 | 0.80 | 0.85 | 0.86 | 0.81 |
| ResNet | 0.79 | 0.84 | 0.7 | 0.81 | 0.43 | 0.59 | **0.63** | 0.78 | 0.84 | 0.83 | 0.74 | **0.83** | 0.68 | 0.52 | **0.61** | 0.76 | 0.59 | 0.62 |
| CNN | 0.0 | 0.0 | 0.0 | 0.0 | 0.01 | 0.28 | 0.0 | 0.0 | 0.0 | 0.0 | 0.0 | 0.0 | 0.0 | 0.0 | 0.0 | 0.0 | **1.0** | 0.39 |
| AWD-LSTM | 0.0 | 0.33 | 0.0 | 0.0 | 0.26 | 0.28 | 0.0 | 0.0 | 0.0 | 0.22 | 0.0 | 0.0 | 0.02 | 0.38 | 0.0 | 0.0 | 0.81 | 0.4 |
| ResNet-LM | 0.59 | 0.62 | 0.0 | 0.0 | 0.52 | 0.41 | 0.0 | 0.0 | 0.51 | 0.76 | 0.0 | 0.0 | 0.56 | 0.46 | 0.0 | 0.0 | 0.43 | 0.55 |
| NT-H | 0.67 | 0.59 | 0.0 | 0.0 | 0.59 | 0.49 | 0.0 | 0.0 | 0.6 | 0.7 | 0.0 | 0.22 | 0.65 | 0.54 | 0.0 | 0.0 | 0.49 | 0.66 |
| NT-MS | **0.94** | 0.89 | 0.73 | 0.66 | **0.84** | **0.69** | 0.5 | 0.73 | 0.93 | 0.89 | 0.64 | 0.74 | **0.86** | **0.69** | 0.57 | 0.66 | 0.55 | **0.79** |
| NT-1000G | 0.78 | 0.79 | 0.03 | 0.28 | 0.7 | 0.59 | 0.01 | 0.64 | 0.76 | 0.84 | 0.14 | 0.62 | 0.74 | 0.63 | 0.06 | 0.43 | 0.57 | 0.7 |
| NT-V2 | **0.94** | **0.91** | **0.75** | 0.73 | 0.78 | 0.65 | 0.55 | **0.8** | **0.94** | **0.91** | **0.75** | 0.74 | 0.81 | 0.68 | 0.59 | **0.77** | 0.57 | 0.77 |
| DNABERT | 0.43 | 0.49 | 0.47 | 0.33 | 0.54 | 0.34 | 0.24 | 0.36 | 0.39 | 0.58 | 0.2 | 0.52 | 0.2 | 0.45 | 0.38 | 0.35 | 0.56 | 0.5 |
| DNABERT2 | 0.51 | 0.69 | 0.09 | 0.42 | 0.6 | 0.5 | 0.0 | 0.0 | 0.51 | 0.69 | 0.13 | 0.49 | 0.57 | 0.56 | 0.0 | 0.0 | 0.63 | 0.65 |
| GENA-LM BERT | 0.82 | 0.81 | 0.34 | **0.83** | 0.69 | 0.6 | 0.29 | 0.59 | 0.82 | 0.81 | 0.26 | 0.57 | 0.7 | 0.61 | 0.31 | 0.65 | 0.53 | 0.65 |
| GENA-LM BigBird | 0.41 | 0.53 | 0.13 | 0.35 | 0.59 | 0.49 | 0.13 | 0.33 | 0.39 | 0.56 | 0.11 | 0.33 | 0.75 | 0.51 | 0.04 | 0.41 | 0.43 | 0.66 |
| HyenaDNA tiny | 0.17 | 0.26 | 0.05 | 0.11 | 0.29 | 0.33 | 0.02 | 0.23 | 0.02 | 0.39 | 0.06 | 0.43 | 0.04 | 0.46 | 0.0 | 0.0 | 0.79 | 0.41 |
| HyenaDNA large | 0.23 | 0.4 | 0.04 | 0.18 | 0.6 | 0.45 | 0.06 | 0.23 | 0.36 | 0.4 | 0.23 | 0.38 | 0.62 | 0.52 | 0.0 | 0.08 | 0.48 | 0.62 |
| GROVER | 0.31 | 0.53 | 0.12 | 0.29 | 0.55 | 0.39 | 0.01 | 0.14 | 0.45 | 0.48 | 0.26 | 0.36 | 0.47 | 0.48 | 0.06 | 0.25 | 0.48 | 0.55 |

**Table A9:** Chromatin accessibility prediction performance per cell line.

| Cell line | Basset | CNN | AWD-LSTM | ResNet-LM | NT-H | NT-MS | NT-1000G (2.5B) | NT-V2 | DNABERT | DNABERT-2 | GENA-LM BERT | GENA-LM BigBird | HyenaDNA tiny | HyenaDNA large | GROVER |
|---|---|---|---|---|---|---|---|---|---|---|---|---|---|---|---|
| 8988T | **0.86** | 0.82 | 0.82 | 0.85 | 0.83 | 0.85 | 0.84 | 0.84 | 0.86 | 0.85 | 0.84 | 0.85 | 0.84 | 0.85 | 0.85 |
| AoSMC | **0.89** | 0.75 | 0.68 | 0.84 | 0.75 | 0.80 | 0.78 | 0.81 | 0.87 | 0.82 | 0.78 | 0.83 | 0.79 | 0.85 | 0.83 |
| Chorion | **0.81** | 0.78 | 0.77 | 0.81 | 0.79 | **0.81** | 0.80 | 0.80 | 0.82 | **0.81** | 0.80 | **0.81** | 0.80 | **0.81** | |
| CLL | **0.87** | 0.81 | 0.80 | 0.86 | 0.82 | 0.84 | 0.83 | 0.85 | 0.88 | 0.86 | 0.83 | 0.86 | 0.84 | 0.86 | 0.86 |
| Fibrobl | 0.71 | 0.68 | 0.67 | 0.71 | 0.69 | 0.71 | 0.69 | 0.70 | **0.72** | 0.71 | 0.70 | 0.71 | 0.70 | 0.71 | 0.71 |
| FibroP | **0.78** | 0.69 | 0.65 | 0.75 | 0.70 | 0.74 | 0.72 | 0.74 | 0.77 | 0.75 | 0.72 | 0.75 | 0.72 | 0.76 | 0.75 |
| Gliobla | **0.86** | 0.75 | 0.71 | 0.84 | 0.77 | 0.82 | 0.80 | 0.82 | **0.86** | 0.83 | 0.80 | 0.83 | 0.81 | 0.85 | 0.83 |
| GM12891 | **0.89** | 0.83 | 0.81 | 0.87 | 0.83 | 0.85 | 0.84 | 0.86 | **0.89** | 0.87 | 0.84 | 0.87 | 0.85 | 0.88 | 0.87 |
| GM12892 | 0.88 | 0.84 | 0.83 | 0.87 | 0.85 | 0.86 | 0.85 | 0.86 | **0.89** | 0.87 | 0.86 | 0.87 | 0.86 | 0.88 | 0.87 |
| GM18507 | **0.87** | 0.77 | 0.74 | 0.84 | 0.77 | 0.80 | 0.78 | 0.82 | **0.87** | 0.83 | 0.79 | 0.84 | 0.80 | 0.85 | 0.84 |
| GM19238 | 0.86 | 0.79 | 0.77 | 0.84 | 0.80 | 0.82 | 0.81 | 0.83 | **0.87** | 0.84 | 0.81 | 0.84 | 0.82 | 0.85 | 0.84 |
| GM19239 | 0.87 | 0.79 | 0.77 | 0.85 | 0.80 | 0.82 | 0.81 | 0.84 | **0.88** | 0.85 | 0.81 | 0.85 | 0.82 | 0.86 | 0.85 |
| GM19240 | 0.81 | 0.74 | 0.73 | 0.80 | 0.76 | 0.77 | 0.76 | 0.78 | **0.82** | 0.79 | 0.77 | 0.80 | 0.77 | 0.80 | 0.79 |
| H9ES | 0.88 | 0.81 | 0.79 | 0.86 | 0.82 | 0.85 | 0.85 | 0.85 | **0.89** | 0.85 | 0.83 | 0.86 | 0.84 | 0.87 | 0.86 |
| HeLa-S3_IFNa4h | **0.85** | 0.72 | 0.70 | 0.82 | 0.76 | 0.81 | 0.79 | 0.81 | **0.85** | 0.81 | 0.79 | 0.82 | 0.79 | 0.83 | 0.82 |
| Hepatocytes | 0.72 | 0.73 | 0.72 | 0.75 | 0.74 | 0.75 | 0.74 | 0.74 | **0.76** | 0.76 | 0.75 | 0.76 | 0.74 | 0.75 | 0.75 |
| HPDE6-E6E7 | **0.90** | 0.75 | 0.70 | 0.85 | 0.77 | 0.83 | 0.81 | 0.84 | 0.88 | 0.84 | 0.81 | 0.85 | 0.82 | 0.87 | 0.85 |
| HSMM_emb | **0.90** | 0.80 | 0.77 | 0.88 | 0.82 | 0.87 | 0.85 | 0.87 | **0.90** | 0.88 | 0.85 | 0.88 | 0.85 | 0.89 | 0.88 |
| HTR8svn | **0.91** | 0.76 | 0.72 | 0.86 | 0.78 | 0.84 | 0.82 | 0.85 | 0.89 | 0.85 | 0.82 | 0.86 | 0.83 | 0.88 | 0.86 |
| Huh-7.5 | 0.81 | 0.76 | 0.75 | 0.81 | 0.78 | 0.80 | 0.79 | 0.80 | **0.83** | 0.81 | 0.79 | 0.81 | 0.79 | 0.82 | 0.81 |
| Huh-7 | 0.84 | 0.77 | 0.75 | 0.83 | 0.78 | 0.81 | 0.80 | 0.81 | **0.86** | 0.82 | 0.80 | 0.82 | 0.80 | 0.84 | 0.83 |
| iPS | **0.91** | 0.87 | 0.87 | 0.90 | 0.88 | 0.90 | 0.89 | 0.90 | **0.91** | 0.90 | 0.88 | 0.90 | 0.89 | **0.91** | 0.90 |
| Ishikawa_Estradiol | **0.85** | 0.76 | 0.74 | 0.83 | 0.78 | 0.81 | 0.79 | 0.81 | **0.85** | 0.81 | 0.78 | 0.82 | 0.79 | 0.84 | 0.82 |
| Ishikawa_4OHTAM | 0.85 | 0.77 | 0.75 | 0.83 | 0.78 | 0.81 | 0.80 | 0.81 | **0.86** | 0.82 | 0.79 | 0.83 | 0.80 | 0.84 | 0.83 |
| LNCaP_androgen | 0.82 | 0.76 | 0.74 | 0.83 | 0.77 | 0.79 | 0.78 | 0.79 | **0.85** | 0.81 | 0.78 | 0.81 | 0.78 | 0.83 | 0.82 |
| MCF-7_Hypoxia | 0.83 | 0.75 | 0.74 | 0.81 | 0.76 | 0.79 | 0.78 | 0.80 | **0.85** | 0.80 | 0.77 | 0.80 | 0.78 | 0.81 | 0.80 |
| Medullo | 0.72 | 0.71 | 0.69 | 0.73 | 0.71 | 0.72 | 0.71 | 0.72 | **0.75** | 0.74 | 0.71 | 0.74 | 0.72 | 0.74 | 0.73 |
| Melano | **0.71** | 0.65 | 0.63 | 0.70 | 0.66 | 0.68 | 0.67 | 0.68 | **0.71** | 0.70 | 0.67 | 0.69 | 0.67 | 0.70 | 0.69 |
| Myometr | **0.84** | 0.74 | 0.68 | 0.82 | 0.75 | 0.80 | 0.78 | 0.80 | **0.84** | 0.81 | 0.77 | 0.81 | 0.79 | 0.83 | 0.81 |
| Osteobl | 0.72 | 0.69 | 0.68 | 0.72 | 0.70 | 0.72 | 0.71 | 0.71 | **0.73** | 0.72 | 0.71 | 0.72 | 0.71 | 0.73 | 0.72 |
| PanIsletD | **0.85** | 0.74 | 0.68 | 0.82 | 0.74 | 0.79 | 0.78 | 0.80 | 0.84 | 0.81 | 0.76 | 0.81 | 0.79 | 0.83 | 0.82 |
| PanIslets | 0.79 | 0.75 | 0.74 | 0.80 | 0.77 | 0.80 | 0.78 | 0.79 | **0.81** | 0.80 | 0.79 | 0.80 | 0.78 | 0.80 | 0.80 |
| pHTE | **0.81** | 0.73 | 0.70 | 0.79 | 0.74 | 0.78 | 0.76 | 0.78 | **0.81** | 0.78 | 0.76 | 0.78 | 0.77 | 0.80 | 0.79 |
| ProgFib | **0.85** | 0.76 | 0.71 | 0.83 | 0.76 | 0.80 | 0.79 | 0.81 | **0.85** | 0.82 | 0.78 | 0.82 | 0.80 | 0.84 | 0.82 |
| RWPE1 | **0.90** | 0.74 | 0.68 | 0.85 | 0.76 | 0.83 | 0.81 | 0.83 | 0.88 | 0.84 | 0.80 | 0.84 | 0.82 | 0.87 | 0.85 |
| Stellate | **0.88** | 0.77 | 0.71 | 0.85 | 0.77 | 0.82 | 0.81 | 0.83 | 0.87 | 0.84 | 0.80 | 0.84 | 0.81 | 0.86 | 0.84 |
| T-47D | **0.81** | 0.75 | 0.73 | 0.79 | 0.75 | 0.78 | 0.77 | 0.78 | **0.81** | 0.79 | 0.77 | 0.79 | 0.76 | 0.80 | 0.79 |
| CD4_Th0 | 0.79 | 0.76 | 0.75 | 0.80 | 0.77 | 0.79 | 0.78 | 0.79 | **0.80** | 0.80 | 0.78 | 0.79 | 0.78 | **0.80** | 0.79 |
| Urothelia | **0.90** | 0.79 | 0.76 | 0.87 | 0.81 | 0.85 | 0.84 | 0.86 | 0.89 | 0.87 | 0.83 | 0.86 | 0.84 | 0.88 | 0.87 |
| Urothelia_UT189 | 0.85 | 0.76 | 0.73 | 0.82 | 0.78 | 0.81 | 0.80 | 0.81 | **0.88** | 0.82 | 0.79 | 0.81 | 0.80 | 0.84 | 0.83 |
| AG04449 | **0.90** | 0.74 | 0.65 | 0.83 | 0.72 | 0.79 | 0.76 | 0.80 | 0.87 | 0.81 | 0.75 | 0.82 | 0.78 | 0.86 | 0.83 |
| AG04450 | **0.89** | 0.74 | 0.66 | 0.84 | 0.74 | 0.80 | 0.78 | 0.81 | 0.87 | 0.82 | 0.77 | 0.83 | 0.79 | 0.86 | 0.83 |
| AG09309 | **0.89** | 0.73 | 0.65 | 0.83 | 0.72 | 0.78 | 0.76 | 0.80 | 0.87 | 0.80 | 0.75 | 0.82 | 0.78 | 0.85 | 0.82 |
| AG09319 | **0.89** | 0.75 | 0.67 | 0.84 | 0.74 | 0.80 | 0.78 | 0.81 | 0.87 | 0.82 | 0.77 | 0.83 | 0.79 | 0.85 | 0.83 |
| AG10803 | **0.90** | 0.74 | 0.65 | 0.83 | 0.72 | 0.78 | 0.76 | 0.80 | 0.87 | 0.81 | 0.75 | 0.82 | 0.78 | 0.86 | 0.83 |
| AoAF | **0.89** | 0.74 | 0.66 | 0.83 | 0.73 | 0.79 | 0.77 | 0.80 | 0.87 | 0.81 | 0.76 | 0.82 | 0.78 | 0.85 | 0.83 |
| BE2_C | 0.80 | 0.73 | 0.69 | 0.80 | 0.72 | 0.76 | 0.75 | 0.77 | 0.83 | 0.78 | 0.73 | 0.79 | 0.75 | **0.81** | 0.79 |
| BJ | **0.89** | 0.75 | 0.66 | 0.83 | 0.73 | 0.79 | 0.77 | 0.80 | 0.87 | 0.81 | 0.76 | 0.82 | 0.78 | 0.85 | 0.83 |
| Caco-2 | 0.91 | 0.91 | 0.90 | 0.92 | 0.91 | 0.91 | 0.91 | 0.91 | 0.93 | 0.92 | 0.90 | **0.92** | 0.91 | **0.92** | 0.93 |
| CD20+ | **0.87** | 0.78 | 0.73 | 0.84 | 0.76 | 0.80 | 0.79 | 0.83 | 0.88 | 0.84 | 0.78 | 0.84 | 0.80 | 0.85 | 0.84 |
| CD34+ | **0.87** | 0.75 | 0.70 | 0.83 | 0.73 | 0.78 | 0.77 | 0.82 | 0.87 | 0.81 | 0.76 | 0.83 | 0.78 | 0.84 | 0.83 |
| CMK | 0.81 | 0.74 | 0.69 | 0.81 | 0.72 | 0.76 | 0.74 | 0.79 | **0.85** | 0.78 | 0.72 | 0.81 | 0.76 | 0.81 | 0.81 |
| GM06990 | 0.85 | 0.78 | 0.72 | 0.83 | 0.73 | 0.75 | 0.75 | 0.80 | **0.86** | 0.81 | 0.74 | 0.82 | 0.78 | 0.83 | 0.82 |
| GM12864 | **0.85** | 0.74 | 0.67 | 0.80 | 0.69 | 0.73 | 0.72 | 0.78 | **0.85** | 0.79 | 0.71 | 0.80 | 0.76 | 0.82 | 0.80 |
| GM12865 | **0.86** | 0.74 | 0.67 | 0.81 | 0.70 | 0.74 | 0.73 | 0.79 | 0.85 | 0.79 | 0.71 | 0.80 | 0.76 | 0.83 | 0.81 |
| H7-hESC | 0.78 | 0.72 | 0.69 | 0.78 | 0.73 | 0.75 | 0.75 | 0.76 | 0.83 | 0.77 | 0.71 | **0.79** | 0.75 | **0.79** | 0.79 |
| HAc | **0.88** | 0.74 | 0.66 | 0.83 | 0.74 | 0.79 | 0.78 | 0.81 | 0.87 | 0.81 | 0.76 | 0.83 | 0.78 | 0.86 | 0.83 |
| HAEpiC | **0.86** | 0.72 | 0.63 | 0.81 | 0.70 | 0.76 | 0.74 | 0.78 | **0.86** | 0.79 | 0.73 | 0.80 | 0.76 | 0.84 | 0.81 |

| Label | Basset | CNN | AWD-LSTM | ResNet-LM | NT-H | NT-MS | NT-1000G (2.5B) | NT-V2 | DNABERT | DNABERT-2 | GENA-LM BERT | GENA-LM BigBird | HyenaDNA tiny | HyenaDNA large | GROVER |
|---|---|---|---|---|---|---|---|---|---|---|---|---|---|---|---|
| HA-h | **0.88** | 0.74 | 0.67 | 0.83 | 0.74 | 0.79 | 0.78 | 0.81 | 0.86 | 0.81 | 0.77 | 0.83 | 0.79 | 0.85 | 0.83 |
| HA-sp | **0.84** | 0.73 | 0.68 | 0.81 | 0.73 | 0.78 | 0.76 | 0.79 | 0.83 | 0.80 | 0.76 | 0.80 | 0.77 | 0.82 | 0.80 |
| HBMEC | **0.89** | 0.73 | 0.64 | 0.83 | 0.72 | 0.80 | 0.78 | 0.82 | 0.87 | 0.82 | 0.77 | 0.83 | 0.79 | 0.86 | 0.83 |
| HCF | **0.89** | 0.74 | 0.66 | 0.83 | 0.73 | 0.79 | 0.77 | 0.80 | 0.87 | 0.81 | 0.76 | 0.83 | 0.78 | 0.86 | 0.83 |
| HCFaa | **0.89** | 0.72 | 0.64 | 0.83 | 0.72 | 0.80 | 0.77 | 0.81 | 0.87 | 0.81 | 0.76 | 0.82 | 0.78 | 0.85 | 0.82 |
| HCM | **0.89** | 0.73 | 0.65 | 0.82 | 0.72 | 0.78 | 0.76 | 0.80 | 0.87 | 0.80 | 0.75 | 0.82 | 0.77 | 0.85 | 0.82 |
| HConF | **0.89** | 0.74 | 0.66 | 0.84 | 0.74 | 0.81 | 0.78 | 0.81 | 0.87 | 0.82 | 0.78 | 0.83 | 0.79 | 0.86 | 0.83 |
| HCPEpiC | **0.88** | 0.71 | 0.64 | 0.81 | 0.71 | 0.77 | 0.75 | 0.79 | 0.85 | 0.79 | 0.74 | 0.80 | 0.76 | 0.84 | 0.81 |
| HCT-116 | **0.89** | 0.73 | 0.68 | 0.85 | 0.75 | 0.84 | 0.81 | 0.85 | 0.88 | 0.84 | 0.80 | 0.85 | 0.81 | 0.87 | 0.85 |
| HEEpiC | **0.90** | 0.71 | 0.62 | 0.82 | 0.71 | 0.78 | 0.76 | 0.80 | 0.87 | 0.80 | 0.74 | 0.81 | 0.78 | 0.85 | 0.81 |
| HFF | **0.89** | 0.73 | 0.65 | 0.83 | 0.73 | 0.79 | 0.77 | 0.80 | 0.86 | 0.81 | 0.76 | 0.82 | 0.78 | 0.85 | 0.82 |
| HFF-Myc | **0.86** | 0.71 | 0.64 | 0.81 | 0.71 | 0.77 | 0.75 | 0.78 | 0.85 | 0.79 | 0.74 | 0.80 | 0.76 | 0.83 | 0.81 |
| HGF | **0.89** | 0.76 | 0.66 | 0.83 | 0.73 | 0.78 | 0.77 | 0.80 | 0.87 | 0.81 | 0.75 | 0.82 | 0.78 | 0.85 | 0.83 |
| HIPEpiC | **0.88** | 0.72 | 0.64 | 0.81 | 0.71 | 0.78 | 0.76 | 0.79 | 0.86 | 0.80 | 0.74 | 0.81 | 0.77 | 0.84 | 0.81 |
| HL-60 | **0.81** | 0.70 | 0.64 | 0.77 | 0.67 | 0.69 | 0.68 | 0.73 | 0.83 | 0.76 | 0.66 | 0.75 | 0.72 | 0.80 | 0.77 |
| HMF | **0.90** | 0.74 | 0.64 | 0.84 | 0.73 | 0.81 | 0.78 | 0.82 | 0.88 | 0.83 | 0.77 | 0.83 | 0.80 | 0.87 | 0.84 |
| HMVEC-dAd | **0.89** | 0.76 | 0.70 | 0.85 | 0.76 | 0.81 | 0.79 | 0.83 | 0.88 | 0.83 | 0.79 | 0.85 | 0.80 | 0.87 | 0.84 |
| HMVEC-dBl-Ad | **0.89** | 0.74 | 0.66 | 0.84 | 0.72 | 0.78 | 0.76 | 0.82 | 0.88 | 0.81 | 0.75 | 0.83 | 0.78 | 0.86 | 0.83 |
| HMVEC-dBl-Neo | **0.88** | 0.73 | 0.66 | 0.82 | 0.72 | 0.78 | 0.76 | 0.81 | 0.86 | 0.80 | 0.75 | 0.82 | 0.77 | 0.85 | 0.82 |
| HMVEC-dLy-Ad | **0.88** | 0.76 | 0.68 | 0.83 | 0.74 | 0.79 | 0.78 | 0.81 | 0.87 | 0.81 | 0.77 | 0.83 | 0.79 | 0.86 | 0.83 |
| HMVEC-dLy-Neo | **0.89** | 0.75 | 0.67 | 0.84 | 0.73 | 0.79 | 0.77 | 0.82 | 0.87 | 0.81 | 0.76 | 0.83 | 0.79 | 0.86 | 0.83 |
| HMVEC-dNeo | **0.89** | 0.76 | 0.69 | 0.84 | 0.75 | 0.80 | 0.78 | 0.82 | 0.88 | 0.82 | 0.77 | 0.84 | 0.79 | 0.86 | 0.84 |
| HMVEC-LBl | **0.89** | 0.73 | 0.65 | 0.83 | 0.72 | 0.79 | 0.77 | 0.82 | 0.87 | 0.81 | 0.76 | 0.83 | 0.78 | 0.86 | 0.83 |
| HMVEC-LLy | **0.87** | 0.75 | 0.68 | 0.82 | 0.74 | 0.78 | 0.77 | 0.80 | 0.86 | 0.80 | 0.76 | 0.82 | 0.78 | 0.85 | 0.82 |
| HNPCEpiC | **0.89** | 0.72 | 0.64 | 0.83 | 0.72 | 0.79 | 0.77 | 0.81 | 0.87 | 0.81 | 0.76 | 0.82 | 0.78 | 0.85 | 0.83 |
| HPAEC | **0.88** | 0.75 | 0.68 | 0.83 | 0.74 | 0.80 | 0.78 | 0.82 | 0.87 | 0.82 | 0.77 | 0.84 | 0.79 | 0.86 | 0.83 |
| HPAF | **0.89** | 0.73 | 0.65 | 0.83 | 0.72 | 0.79 | 0.77 | 0.80 | 0.87 | 0.81 | 0.75 | 0.82 | 0.78 | 0.86 | 0.83 |
| HPdLF | **0.89** | 0.75 | 0.66 | 0.83 | 0.73 | 0.79 | 0.77 | 0.80 | 0.86 | 0.81 | 0.76 | 0.82 | 0.78 | 0.85 | 0.83 |
| HPF | **0.90** | 0.75 | 0.67 | 0.85 | 0.75 | 0.81 | 0.79 | 0.82 | 0.88 | 0.83 | 0.78 | 0.83 | 0.79 | 0.87 | 0.84 |
| HRCEpiC | 0.85 | 0.72 | 0.65 | 0.82 | 0.72 | 0.78 | 0.76 | 0.79 | **0.85** | 0.79 | 0.75 | 0.81 | 0.78 | 0.84 | 0.81 |
| HRE | 0.87 | 0.73 | 0.65 | 0.83 | 0.73 | 0.80 | 0.78 | 0.81 | **0.87** | 0.82 | 0.77 | 0.83 | 0.80 | 0.85 | 0.83 |
| HRGEC | **0.88** | 0.73 | 0.66 | 0.83 | 0.73 | 0.79 | 0.77 | 0.82 | 0.86 | 0.81 | 0.76 | 0.83 | 0.78 | 0.85 | 0.83 |
| HRPEpiC | 0.83 | 0.73 | 0.65 | 0.80 | 0.70 | 0.75 | 0.74 | 0.77 | **0.84** | 0.79 | 0.73 | 0.79 | 0.76 | 0.83 | 0.80 |
| HVMF | **0.86** | 0.74 | 0.65 | 0.82 | 0.71 | 0.77 | 0.75 | 0.78 | 0.85 | 0.80 | 0.74 | 0.80 | 0.76 | 0.83 | 0.81 |
| Jurkat | 0.82 | 0.72 | 0.65 | 0.80 | 0.67 | 0.71 | 0.70 | 0.78 | **0.84** | 0.76 | 0.68 | 0.78 | 0.74 | 0.82 | 0.79 |
| Monocytes-CD14+ | 0.86 | 0.74 | 0.67 | 0.82 | 0.71 | 0.75 | 0.74 | 0.80 | **0.88** | 0.81 | 0.72 | 0.81 | 0.77 | 0.84 | 0.82 |
| NB4 | 0.87 | 0.74 | 0.68 | 0.83 | 0.72 | 0.77 | 0.75 | 0.80 | **0.88** | 0.81 | 0.74 | 0.82 | 0.77 | 0.85 | 0.83 |
| NH-A | **0.89** | 0.75 | 0.66 | 0.84 | 0.73 | 0.79 | 0.77 | 0.81 | 0.87 | 0.82 | 0.76 | 0.83 | 0.80 | 0.86 | 0.84 |
| NHDF-Ad | **0.87** | 0.74 | 0.64 | 0.81 | 0.70 | 0.76 | 0.75 | 0.78 | 0.85 | 0.79 | 0.73 | 0.80 | 0.76 | 0.84 | 0.81 |
| NHDF-neo | **0.87** | 0.76 | 0.65 | 0.82 | 0.71 | 0.77 | 0.76 | 0.79 | 0.86 | 0.80 | 0.74 | 0.81 | 0.77 | 0.84 | 0.82 |
| NHLF | **0.89** | 0.74 | 0.65 | 0.83 | 0.73 | 0.79 | 0.77 | 0.81 | 0.87 | 0.81 | 0.76 | 0.83 | 0.79 | 0.86 | 0.83 |
| NT2-D1 | 0.82 | 0.74 | 0.71 | 0.81 | 0.76 | 0.79 | 0.78 | 0.80 | **0.85** | 0.80 | 0.76 | 0.82 | 0.78 | 0.82 | 0.81 |
| PANC-1 | **0.86** | 0.71 | 0.65 | 0.82 | 0.73 | 0.81 | 0.79 | 0.82 | 0.85 | 0.81 | 0.78 | 0.82 | 0.78 | 0.84 | 0.82 |
| PrEC | **0.89** | 0.73 | 0.64 | 0.83 | 0.72 | 0.79 | 0.77 | 0.80 | 0.87 | 0.80 | 0.75 | 0.82 | 0.79 | 0.85 | 0.82 |
| RPTEC | **0.84** | 0.71 | 0.65 | 0.81 | 0.72 | 0.78 | 0.75 | 0.79 | **0.84** | 0.79 | 0.74 | 0.80 | 0.77 | 0.83 | 0.80 |
| SAEC | **0.90** | 0.71 | 0.62 | 0.82 | 0.71 | 0.79 | 0.77 | 0.80 | 0.87 | 0.80 | 0.75 | 0.81 | 0.78 | 0.85 | 0.82 |
| SKMC | **0.88** | 0.73 | 0.65 | 0.83 | 0.73 | 0.78 | 0.76 | 0.79 | 0.87 | 0.81 | 0.75 | 0.81 | 0.78 | 0.85 | 0.82 |
| SK-N-MC | **0.81** | 0.74 | 0.68 | 0.79 | 0.71 | 0.76 | 0.74 | 0.77 | **0.81** | 0.77 | 0.73 | 0.78 | 0.75 | 0.80 | 0.78 |
| SK-N-SH_RA | 0.87 | 0.85 | 0.81 | 0.89 | 0.83 | 0.85 | 0.85 | 0.86 | **0.90** | 0.87 | 0.83 | 0.88 | 0.85 | 0.89 | 0.88 |
| Th2 | 0.86 | 0.79 | 0.73 | 0.84 | 0.76 | 0.78 | 0.78 | 0.83 | **0.87** | 0.82 | 0.77 | 0.84 | 0.81 | 0.86 | 0.84 |
| WERI-Rb-1 | 0.75 | 0.75 | 0.65 | 0.81 | 0.70 | 0.70 | 0.72 | 0.77 | **0.86** | 0.79 | 0.67 | 0.79 | 0.75 | 0.82 | 0.80 |
| WI-38 | **0.89** | 0.73 | 0.64 | 0.83 | 0.72 | 0.79 | 0.77 | 0.81 | 0.87 | 0.81 | 0.76 | 0.82 | 0.78 | 0.85 | 0.83 |
| WI-38_4OHTAM | **0.84** | 0.72 | 0.62 | 0.81 | 0.70 | 0.77 | 0.75 | 0.79 | **0.84** | 0.79 | 0.74 | 0.80 | 0.77 | 0.83 | 0.80 |
| A549 | **0.84** | 0.71 | 0.67 | 0.81 | 0.74 | 0.79 | 0.78 | 0.80 | **0.84** | 0.80 | 0.77 | 0.80 | 0.78 | 0.82 | 0.81 |
| GM12878 | **0.82** | 0.73 | 0.69 | 0.78 | 0.71 | 0.74 | 0.73 | 0.77 | **0.82** | 0.78 | 0.73 | 0.78 | 0.75 | 0.80 | 0.78 |
| H1-hESC | 0.86 | 0.82 | 0.80 | 0.85 | 0.82 | 0.84 | 0.84 | 0.84 | **0.87** | 0.84 | 0.82 | 0.85 | 0.84 | 0.86 | 0.85 |
| HeLa-S3 | 0.82 | 0.70 | 0.66 | 0.79 | 0.71 | 0.76 | 0.75 | 0.77 | **0.82** | 0.78 | 0.74 | 0.78 | 0.76 | 0.80 | 0.78 |
| HepG2 | 0.85 | 0.79 | 0.78 | 0.84 | 0.80 | 0.83 | 0.82 | 0.83 | **0.86** | 0.84 | 0.81 | 0.84 | 0.82 | 0.85 | 0.84 |
| HMEC | **0.80** | 0.71 | 0.68 | 0.77 | 0.72 | 0.76 | 0.74 | 0.76 | 0.79 | 0.76 | 0.73 | 0.77 | 0.75 | 0.78 | 0.77 |
| HSMM | **0.84** | 0.72 | 0.65 | 0.80 | 0.71 | 0.75 | 0.74 | 0.77 | 0.83 | 0.78 | 0.73 | 0.79 | 0.76 | 0.81 | 0.80 |
| HSMMtube | 0.83 | 0.74 | 0.69 | 0.80 | 0.73 | 0.77 | 0.76 | 0.78 | **0.84** | 0.79 | 0.74 | 0.79 | 0.77 | 0.82 | 0.80 |
| HUVEC | **0.86** | 0.75 | 0.69 | 0.83 | 0.75 | 0.79 | 0.78 | 0.81 | 0.85 | 0.81 | 0.77 | 0.82 | 0.79 | 0.84 | 0.82 |
| K562 | 0.76 | 0.73 | 0.69 | 0.78 | 0.71 | 0.74 | 0.73 | 0.75 | **0.81** | 0.75 | 0.72 | 0.77 | 0.74 | 0.77 | 0.78 |
| LNCaP | 0.74 | 0.71 | 0.67 | 0.75 | 0.68 | 0.70 | 0.69 | 0.71 | **0.77** | 0.73 | 0.68 | 0.73 | 0.70 | 0.76 | 0.74 |
| MCF-7 | **0.80** | 0.69 | 0.67 | 0.77 | 0.69 | 0.73 | 0.72 | 0.74 | 0.79 | 0.75 | 0.71 | 0.75 | 0.72 | 0.77 | 0.76 |
| NHEK | **0.86** | 0.72 | 0.67 | 0.81 | 0.74 | 0.79 | 0.77 | 0.80 | 0.85 | 0.80 | 0.76 | 0.81 | 0.78 | 0.83 | 0.81 |
| Th1 | 0.77 | 0.75 | 0.74 | 0.78 | 0.76 | 0.77 | 0.76 | 0.77 | **0.78** | 0.78 | 0.76 | 0.78 | 0.77 | **0.78** | 0.78 |

**Table A10:** Histone modification prediction performance per label.

| Modification (label no.) | Basset | CNN | AWD-LSTM | ResNet-LM | NT-H | NT-MS | NT-1000G (2.5B) | NT-V2 | DNABERT | DNABERT-2 | GENA-LM BERT | GENA-LM BigBird | HyenaDNA tiny | HyenaDNA large | GROVER |
|---|---|---|---|---|---|---|---|---|---|---|---|---|---|---|---|
| H3K27me3_K562 (0) | 0.63 | 0.67 | 0.66 | 0.70 | 0.69 | 0.70 | 0.69 | 0.68 | **0.72** | 0.70 | 0.70 | 0.71 | 0.68 | 0.67 | 0.70 |
| H3K9ac_K562 (1) | **0.87** | 0.85 | 0.85 | 0.87 | 0.86 | 0.86 | 0.87 | 0.86 | 0.87 | 0.86 | 0.86 | 0.87 | 0.87 | 0.87 | 0.87 |
| H3K9me3_K562 (2) | 0.74 | 0.77 | 0.75 | 0.84 | 0.83 | 0.83 | 0.83 | 0.78 | 0.83 | 0.84 | **0.86** | 0.86 | 0.79 | 0.80 | 0.82 |
| H3K4me1_K562 (3) | 0.65 | 0.67 | 0.65 | 0.68 | 0.67 | 0.68 | 0.67 | 0.67 | **0.71** | 0.69 | 0.67 | 0.69 | 0.67 | 0.67 | 0.69 |
| H3K9ac_K562 (4) | 0.74 | 0.75 | 0.70 | 0.74 | 0.73 | 0.75 | 0.74 | 0.74 | **0.77** | 0.75 | 0.75 | 0.75 | 0.74 | 0.74 | 0.75 |
| H3K4me1_K562 (5) | 0.80 | 0.80 | 0.80 | 0.81 | 0.81 | 0.81 | 0.81 | 0.80 | **0.82** | 0.81 | 0.80 | 0.81 | 0.81 | 0.81 | 0.81 |
| H3K36me3_K562 (6) | 0.63 | 0.65 | 0.62 | 0.70 | 0.70 | **0.74** | 0.71 | 0.66 | 0.70 | 0.72 | 0.73 | 0.74 | 0.65 | 0.67 | 0.69 |
| H3K36me3_K562 (7) | 0.75 | 0.77 | 0.75 | 0.78 | 0.77 | **0.79** | 0.77 | 0.77 | 0.78 | 0.78 | 0.78 | 0.79 | 0.76 | 0.77 | 0.77 |
| H4K20me1_K562 (8) | 0.62 | 0.69 | 0.69 | 0.71 | 0.69 | 0.71 | 0.69 | 0.69 | **0.72** | 0.71 | 0.70 | 0.72 | 0.69 | 0.69 | 0.71 |

| | | | | | | | | | | | | | | | |
|---|---|---|---|---|---|---|---|---|---|---|---|---|---|---|---|
| H3K27me3_K562 (9) | 0.74 | 0.74 | 0.75 | **0.80** | 0.79 | **0.80** | **0.80** | 0.79 | **0.80** | 0.79 | **0.80** | **0.80** | 0.78 | 0.77 | **0.80** |
| H3K4me3_K562 (10) | 0.88 | 0.89 | 0.87 | 0.89 | 0.88 | 0.89 | 0.89 | 0.89 | **0.9** | 0.89 | 0.89 | 0.89 | 0.89 | 0.89 | 0.89 |
| H3K4me3_K562 (11) | 0.89 | 0.89 | 0.87 | 0.89 | 0.89 | 0.89 | 0.89 | 0.89 | **0.9** | 0.89 | 0.89 | 0.89 | 0.89 | 0.89 | 0.89 |
| H3K4me3_K562 (12) | 0.84 | 0.85 | 0.82 | 0.85 | 0.84 | 0.85 | 0.85 | 0.84 | **0.86** | 0.85 | 0.85 | 0.85 | 0.85 | 0.85 | 0.85 |
| H3K4me3_K562 (13) | 0.76 | 0.77 | 0.72 | 0.77 | 0.75 | 0.77 | 0.76 | 0.76 | **0.80** | 0.77 | 0.77 | 0.78 | 0.76 | 0.75 | 0.77 |
| H3K79me2_K562 (14) | 0.74 | 0.76 | 0.75 | 0.76 | 0.76 | 0.76 | 0.76 | 0.75 | 0.76 | 0.76 | 0.76 | **0.77** | 0.76 | 0.70 | 0.76 |
| H3K4me2_K562 (15) | 0.70 | 0.72 | 0.67 | 0.71 | 0.70 | 0.72 | 0.71 | 0.71 | **0.75** | 0.73 | 0.71 | 0.72 | 0.70 | 0.70 | 0.72 |
| H3K27ac_K562 (16) | 0.70 | 0.72 | 0.67 | 0.71 | 0.70 | 0.72 | 0.70 | 0.71 | **0.76** | 0.73 | 0.71 | 0.72 | 0.71 | 0.71 | 0.72 |
| H2AFZ_K562 (17) | 0.70 | 0.71 | 0.67 | 0.72 | 0.71 | 0.73 | 0.71 | 0.71 | **0.75** | 0.73 | 0.73 | 0.73 | 0.70 | 0.70 | 0.72 |

**Table A11:** CpG methylation prediction performance per cell line.

| Model | SK-N-SH
ENCFF567KCL | GM23248
ENCFF170XYJ | A549
ENCFF948WVD | HepG2
ENCFF690FNR | HUES64
ENCFF890GMD | GM23248
ENCFF840XVU | HeLa-S3
ENCFF754RAW |
|---|---|---|---|---|---|---|---|
| **Basset** | 0.93 | 0.94 | 0.93 | 0.90 | 0.95 | 0.94 | 0.93 |
| **CNN** | 0.84 | 0.84 | 0.84 | 0.82 | 0.93 | 0.84 | 0.83 |
| **ResNet-LM** | 0.86 | 0.87 | 0.86 | 0.85 | 0.94 | 0.87 | 0.86 |
| **AWD-LSTM** | 0.80 | 0.80 | 0.80 | 0.78 | 0.89 | 0.80 | 0.79 |
| **NT-H** | 0.87 | 0.87 | 0.87 | 0.85 | 0.94 | 0.87 | 0.87 |
| **NT-MS** | **0.92** | **0.92** | **0.92** | **0.89** | **0.96** | **0.92** | **0.91** |
| **NT-1000G (2.5B)** | 0.88 | 0.88 | 0.88 | 0.86 | 0.94 | 0.88 | 0.87 |
| **NT-V2** | 0.90 | 0.91 | 0.90 | 0.88 | **0.96** | 0.91 | 0.90 |
| **DNABERT** | 0.91 | 0.91 | 0.91 | 0.88 | **0.96** | 0.91 | 0.90 |
| **DNABERT-2** | 0.89 | 0.89 | 0.89 | 0.87 | 0.96 | 0.89 | 0.89 |
| **GENA-LM BERT** | 0.91 | 0.91 | 0.91 | 0.89 | 0.95 | 0.91 | 0.90 |
| **GENA-LM BigBird** | 0.90 | 0.91 | 0.90 | 0.88 | 0.95 | 0.91 | 0.90 |
| **HyenaDNA tiny** | 0.85 | 0.85 | 0.85 | 0.83 | 0.92 | 0.85 | 0.84 |
| **HyenaDNA large** | 0.91 | 0.91 | 0.91 | 0.88 | 0.94 | 0.91 | 0.90 |
| **GROVER** | 0.88 | 0.89 | 0.88 | 0.86 | 0.94 | 0.89 | 0.88 |

**Table A12:** Variant effect prediction performance (AUROC) on the expression variant effect prediction dataset, stratified by variant category. Categories that only have samples of one label were ommitted as no AUC can be determined. For completeness, also AUROCs on categories with very low sample numbers are reported, but should be interpreted with caution.

| Model | Intron (n=60,072) | Intergenic (n=23,218) | Upstream gene (n=6,339) | Downstream gene (n=4,581) | Regulatory region (n=4,010) | Noncoding transcript exon (n=3,099) | 3' UTR (n=2,025) | 5' UTR (n=462) | TF binding site (n=433) | Splice region (n=129) | Splice polypyrimidine tract (n=103) | Missense (n=60) | Splice donor region (n=31) | Synonymous (n=24) | Splice donor (n=15) | Splice donor 5th base (n=13) | Stop lost (n=2) |
|---|---|---|---|---|---|---|---|---|---|---|---|---|---|---|---|---|---|
| **DeepSEA** | 0.70 | 0.69 | 0.71 | 0.71 | 0.71 | 0.68 | 0.64 | 0.72 | 0.64 | 0.55 | 0.62 | 0.72 | 0.46 | 0.54 | 0.35 | 0.83 | 1.00 |
| **ResNet-LM** | 0.55 | 0.54 | 0.56 | 0.55 | 0.52 | 0.51 | 0.54 | 0.44 | 0.46 | 0.44 | 0.54 | 0.49 | 0.69 | 0.70 | 0.50 | 0.50 | 1.00 |
| **AWD-LSTM** | 0.53 | 0.54 | 0.52 | 0.55 | 0.56 | 0.53 | 0.51 | 0.51 | 0.51 | 0.48 | 0.51 | 0.44 | 0.31 | 0.28 | 0.38 | 0.37 | 0.00 |
| **NT-H** | 0.55 | 0.54 | 0.54 | 0.55 | 0.52 | 0.51 | 0.49 | 0.43 | 0.44 | 0.51 | 0.33 | 0.57 | 0.36 | 0.71 | 0.50 | 0.67 | 1.00 |
| **NT-MS** | 0.55 | 0.53 | 0.54 | 0.55 | 0.54 | 0.55 | **0.57** | 0.48 | 0.53 | 0.54 | 0.51 | 0.54 | 0.56 | 0.65 | 0.19 | 0.60 | 1.00 |
| **NT-1000G-2.5B** | 0.44 | 0.43 | 0.43 | 0.44 | 0.48 | 0.46 | 0.48 | 0.44 | 0.47 | 0.42 | 0.40 | 0.44 | 0.39 | 0.54 | 0.27 | 0.21 | 1.00 |
| **NT-1000G-500M** | 0.49 | 0.48 | 0.49 | 0.47 | 0.50 | 0.53 | 0.50 | 0.45 | 0.51 | 0.51 | 0.48 | 0.40 | 0.66 | 0.29 | 0.46 | 0.33 | 0.00 |
| **NT-V2-500M** | 0.48 | 0.47 | 0.46 | 0.48 | 0.50 | 0.48 | 0.50 | 0.41 | 0.51 | **0.58** | 0.54 | 0.34 | 0.51 | 0.68 | **0.77** | 0.40 | 0.00 |
| **DNABERT** | **0.60** | **0.59** | **0.61** | **0.60** | 0.57 | **0.57** | 0.55 | 0.60 | 0.51 | 0.51 | **0.70** | 0.57 | **0.79** | **0.81** | 0.65 | 0.5 | 0.00 |
| **DNABERT-2** | 0.49 | 0.49 | 0.47 | 0.49 | 0.53 | 0.48 | 0.48 | **0.52** | 0.57 | 0.49 | 0.47 | 0.55 | 0.78 | 0.59 | 0.35 | 0.52 | 1.00 |
| **GENA-LM BERT** | 0.49 | 0.49 | 0.50 | 0.50 | 0.54 | 0.51 | 0.51 | 0.49 | 0.55 | 0.51 | 0.47 | 0.53 | 0.27 | 0.29 | 0.58 | 0.60 | 0.00 |
| **GENA-LM BigBird** | 0.49 | 0.48 | 0.48 | 0.49 | 0.52 | 0.51 | 0.52 | 0.49 | 0.53 | 0.51 | 0.47 | 0.53 | 0.24 | 0.43 | 0.35 | 0.55 | 0.00 |
| **HyenaDNA large** | 0.51 | 0.52 | 0.50 | 0.52 | 0.53 | 0.51 | 0.50 | 0.49 | 0.48 | 0.47 | 0.54 | 0.49 | 0.37 | 0.26 | 0.19 | 0.33 | 0.00 |
| **HyenaDNA medium (160k)** | 0.48 | 0.49 | 0.47 | 0.50 | 0.52 | 0.49 | 0.49 | 0.46 | 0.46 | 0.49 | 0.53 | 0.51 | 0.34 | 0.23 | 0.19 | 0.33 | 0.00 |
| **HyenaDNA medium (450k)** | 0.50 | 0.51 | 0.49 | 0.52 | 0.54 | 0.50 | 0.50 | 0.47 | 0.45 | 0.48 | 0.53 | 0.54 | 0.39 | 0.26 | 0.19 | 0.38 | 0.00 |
| **HyenaDNA small** | 0.46 | 0.47 | 0.45 | 0.47 | 0.50 | 0.47 | 0.48 | 0.47 | 0.50 | 0.49 | 0.50 | 0.42 | 0.32 | 0.25 | 0.19 | 0.33 | 0.00 |
| **HyenaDNA tiny** | 0.47 | 0.48 | 0.44 | 0.49 | 0.51 | 0.49 | 0.48 | 0.45 | 0.50 | 0.48 | 0.49 | 0.39 | 0.37 | 0.35 | 0.23 | 0.24 | 0.00 |
| **GROVER** | 0.55 | 0.55 | 0.58 | 0.55 | 0.55 | 0.56 | 0.56 | 0.50 | 0.56 | 0.41 | 0.48 | **0.66** | 0.36 | 0.46 | 0.42 | **0.74** | 0.00 |

**Table A13:** Variant effect prediction performance (AUROC) on the disease variant effect prediction dataset, stratified by variant category. Categories that only have samples of one label were ommitted as no AUC can be determined. For completeness, also AUCs on categories with very low sample numbers are reported, but should be interpreted with caution.

| Model | Intron (n=138,211) | Splice region (n=40,360) | Splice polypyrimidine tract (n=39,686) | Noncoding transcript exon (23,721) | 3' UTR (n=20,441) | Splice donor (n=10,811) | Splice acceptor (n=9,280) | 5' UTR (n=6,996) | Upstream gene (n=2,264) | Splice donor region (n=2,056) | Splice donor 5th base (n=1,060) | Downstream Gene (n=274) | Mature miRNA (n=40) | Intergenic (n=20) |
|---|---|---|---|---|---|---|---|---|---|---|---|---|---|---|
| DeepSEA | 0.48 | 0.44 | 0.46 | 0.73 | 0.69 | 0.47 | 0.48 | 0.58 | 0.61 | 0.45 | 0.41 | 0.72 | 0.18 | 0.92 |
| ResNet-LM | 0.51 | 0.67 | 0.53 | 0.50 | 0.51 | 0.55 | 0.48 | 0.46 | **0.58** | 0.64 | 0.63 | 0.31 | 0.79 | 0.05 |
| AWD-LSTM | 0.53 | 0.56 | 0.59 | 0.52 | 0.50 | 0.48 | 0.52 | 0.46 | 0.47 | 0.50 | 0.43 | 0.40 | 0.12 | 0.53 |
| NT-H | 0.43 | 0.53 | 0.49 | 0.51 | 0.56 | 0.51 | 0.52 | 0.53 | 0.52 | 0.49 | 0.50 | 0.53 | 0.38 | 0.58 |
| NT-MS | **0.62** | **0.70** | **0.65** | **0.57** | 0.55 | **0.74** | **0.61** | 0.57 | 0.56 | **0.76** | **0.76** | 0.44 | 0.82 | 0.63 |
| NT-1000G-2.5B | 0.49 | 0.57 | 0.54 | 0.52 | 0.48 | 0.51 | 0.50 | 0.47 | 0.45 | 0.52 | 0.52 | 0.45 | 0.10 | 0.11 |
| NT-1000G-500M | 0.46 | 0.53 | 0.50 | 0.49 | 0.40 | 0.51 | 0.49 | 0.47 | 0.41 | 0.47 | 0.49 | 0.36 | 0.03 | 0.63 |
| NT-V2-500M | 0.50 | 0.52 | 0.49 | 0.50 | 0.36 | 0.49 | 0.53 | 0.52 | 0.46 | 0.50 | 0.43 | 0.54 | 0.33 | 0.21 |
| DNABERT | 0.52 | 0.55 | 0.47 | 0.48 | **0.63** | 0.54 | 0.51 | **0.62** | **0.58** | 0.55 | 0.56 | **0.62** | 0.72 | 0.05 |
| DNABERT-2 | 0.48 | 0.46 | 0.53 | 0.54 | 0.49 | 0.50 | 0.52 | 0.45 | 0.51 | 0.45 | 0.52 | 0.51 | 0.92 | 0.95 |
| GENA-LM BERT | 0.50 | 0.49 | 0.48 | 0.51 | 0.50 | 0.56 | 0.60 | 0.50 | 0.41 | 0.47 | 0.42 | 0.50 | 0.95 | **1.00** |
| GENA-LM BigBird | 0.48 | 0.48 | 0.46 | 0.51 | 0.52 | 0.54 | 0.60 | 0.46 | 0.45 | 0.48 | 0.43 | 0.60 | **1.00** | 0.89 |
| HyenaDNA large | 0.53 | 0.52 | 0.59 | 0.52 | 0.48 | 0.44 | 0.52 | 0.48 | 0.48 | 0.48 | 0.42 | 0.40 | 0.00 | 0.63 |
| HyenaDNA medium 160k | 0.52 | 0.54 | 0.60 | 0.54 | 0.46 | 0.44 | 0.51 | 0.46 | 0.50 | 0.47 | 0.41 | 0.41 | 0.00 | 0.47 |
| HyenaDNA medium 450k | 0.53 | 0.53 | 0.58 | 0.51 | 0.46 | 0.45 | 0.50 | 0.51 | 0.50 | 0.47 | 0.42 | 0.35 | 0.00 | 0.37 |
| HyenaDNA small | 0.53 | 0.54 | 0.59 | 0.53 | 0.47 | 0.44 | 0.49 | 0.46 | 0.53 | 0.48 | 0.41 | 0.44 | 0.10 | 0.42 |
| HyenaDNA tiny | 0.53 | 0.55 | 0.59 | 0.53 | 0.47 | 0.44 | 0.52 | 0.48 | 0.50 | 0.48 | 0.42 | 0.44 | 0.08 | 0.37 |
| GROVER | 0.50 | 0.46 | 0.42 | 0.48 | 0.54 | 0.49 | 0.53 | 0.55 | 0.52 | 0.49 | 0,45 | 0.42 | 0.21 | 0.21 |

**Table A14:** Variant effect prediction performance on the disease variant effects prediction dataset with more stringent filtering. Variants labeled as "Likely" in ClinVar were omitted, yielding a reduced dataset (Benign n=100,623, Pathogenic n=8,188). Similarly to the results on the full dataset, NT-MS outperforms DeepSEA. Additionally, ResNet-LM and DNABERT show strong performance.

| Model | AUC |
|---|---|
| DeepSEA | 0.57 |
| ResNet-LM | 0.61 |
| AWD-LSTM | 0.45 |
| NT-H | 0.52 |
| NT-MS | **0.74** |
| NT-1000G-2.5B | 0.49 |
| NT-1000G-500M | 0.46 |
| NT-V2-500M | 0.48 |
| DNABERT | 0.62 |
| DNABERT2 | 0.50 |
| GENA-LM BERT | 0.56 |
| GENA-LM BigBird | 0.52 |
| HyenaDNA large | 0.44 |
| HyenaDNA medium 160k | 0.43 |
| HyenaDNA medium 450k | 0.44 |
| HyenaDNA small | 0.41 |
| HyenaDNA tiny | 0.43 |
| GROVER | 0.52 |

