# OpenReview forum: "BEND: Benchmarking DNA Language Models on Biologically Meaningful Tasks"
_ICLR.cc/2024/Conference — ICLR 2024 poster_

### Official Review · Reviewer_gFhh · 2023-10-28

**Soundness:** 3 good
**Presentation:** 3 good
**Contribution:** 2 fair
**Rating:** 3
**Confidence:** 4

**Summary:**

This paper is concerned with BEND, a Benchmark for DNA language
models, investigating several biologically down-stream tasks defined on the human genome.

It is a interesting to read, as it compares a number of frameworks on biological task of interest. The training data seems to mainly be human curated biological data sets. Is this correct? Although it should not be considered an important critique of the submission, I strongly believe that the future of computational biology will be analysis of more complex data sets having a more complete signal. That is, the annotated data is often not sufficient, and richer experimental datasets will facilitate more interesting analyses. The underlying reason is that the biological reality is more complex: there may be multiple start sites, etc. Nevertheless, the described models can certainly serve as a step in this direction

However, the paper contains quite a lot of background and it is a benchmarking study of several methods that only available as rxiv papers, according to the references. The call for papers does not spell out how focused the conference is on original research, which makes the fit of the paper  somewhat unclear. I would, however, not priorities this paper in a comparison with the average ICLR paper.

**Strengths:**

The paper provides an interesting comparison of the performance of several frameworks across 5-6 prediction task of relevance to modern biology.

**Weaknesses:**

This is not original research and a substantial fraction of the paper is concerned with background material.

**Questions:**

What is the performance of non LM models on these tasks?

---

> ### Author Response · Authors · 2023-11-16
> **First reponse to reviewer gFhh (1/1)**
>
> We thank the reviewer for their excellent comments. We have addressed them here and updated the manuscript to reflect it.
>
> > It is an interesting to read, as it compares a number of frameworks on biological task of interest. The training data seems to mainly be human curated biological data sets. Is this correct? Although it should not be considered an important critique of the submission, I strongly believe that the future of computational biology will be analysis of more complex data sets having a more complete signal. That is, the annotated data is often not sufficient, and richer experimental datasets (...)
>
> Thank you for voicing this concern, we agree that tasks should be as rich as allowed for by experimental data! Our aim was to as far as possible establish BEND based on ground truth experimental evidence, rather than human expert opinion. While human curation can play a role in the underlying data, it is not the case that mostly human-curated data points were used. Rather, decisions were taken based on the availability of experimental data. To briefly summarize:
> * Chromatin accessibility, Histone modification, CpG methylation: the labeled benchmark data was processed from genome-wide experimental raw data available through ENCODE, covering the genome in all its complexity.
> Enhancer annotation: The included enhancers are based on experimental CRISPR interference results, and not manually selected.
> * Gene finding: In order to only include transcripts that have experimental evidence, we applied the following filters to GENCODE: Each included data point needs to (A) be a complete protein coding transcript reviewed by the HAVANA expert group, and (B) have a GENCODE transcript support level of 1 (80% of the benchmark dataset) or 2 (remaining 20%), meaning that there is experimental evidence for the transcript either directly in the GENCODE experimental pipeline or from other sources and that the transcript was reviewed by HAVANA.
> * Variant effect prediction: expression variants (eQTL) are derived from GWAS studies, with individual data points not undergoing human curation to our best knowledge. Disease variants, given that they are obtained from ClinVar, which is based on clinical observations in human patients, undergo a certain level of curation by nature of the data source.
>
> > However, the paper contains quite a lot of background and it is a benchmarking study of several methods that only available as rxiv papers, according to the references. The call for papers does not spell out how focused the conference is on original research, which makes the fit of the paper somewhat unclear. I would, however, not priorities this paper in a comparison with the average ICLR paper.
>
> It is true that the call for papers does not explicitly spell this out, but ICLR does welcome papers with a benchmark focus, with the primary area of this submission being “datasets and benchmarks”. For reference, we would like to highlight a few ICLR papers from previous years.
> * Long Range Arena (Tay, ICLR 2021) -> a collection of long-range prediction tasks to evaluate long-range scaling of different transformer architectures.
> * GeneDisco (Mehrjou, iCLR 2022) -> a collection of active learning tasks for drug discovery.
> * MEDFAIR (Zong, ICLR 2023) -> a collection of benchmark datasets, algorithms and metrics for fairness in medical imaging.
>
> We would therefore like to invite you to consider evaluating our submission as a datasets and benchmarks paper.
> Regarding the arxiv availability - we appreciate your concern. We think that this is still a nascent field and likely to undergo dynamic development over the coming months, making BEND a timely contribution. E.g. HyenaDNA was now accepted at NeurIPS 2023, GPN published in PNAS, DNABERT-2 is under review at this venue, and Nucleotide Transformer’s preprint was updated with additional models in September (now also included in the updated version of our experiments).
> # Weaknesses
> > This is not original research and a substantial fraction of the paper is concerned with background material.
>
> As outlined above, we consider our submission well within what is welcomed by ICLR, and would appreciate your evaluation and feedback accordingly.
>
> We are convinced that the experiments, materials and findings provided by BEND constitute original research, and are not a mere review of existing materials.
>
> # Questions
> > What is the performance of non LM models on these tasks?
>
> Thank you for this question - including the performance of non LM models in the benchmark is of key interest to us, as we think it is absolutely necessary to put LM performance into context of existing non-LM approaches. We have made changes to the “Supervised baselines” section to highlight that the baseline CNN on one-hot encoded DNA does not use DNA LMs, and that all expert methods that we evaluated on the benchmark data do not use DNA LMs either.

---

### Official Review · Reviewer_vBTw · 2023-10-29

**Soundness:** 3 good
**Presentation:** 4 excellent
**Contribution:** 3 good
**Rating:** 6
**Confidence:** 4

**Summary:**

The mansucript presents a set of benchmark datasets for evaluating DNA language models (DNA LMs) and establishes baseline performance for these datasets using existing DNA LMs.  It also compares the LM-based approach to a simple direct supervised learning approach.

**Strengths:**

Strengths:

- While there exist other recent benchmark datasets for DNA LMs, the authors' datasets are complementary to the existing ones.  In particular, the tasks are more realistic than those in the "Genomic benchmarks" paper (but see some caveats below).

- The authors provide a comprehensive comparison of existing DNA LM performance on the benchmark datasets; they also provide a comparison to a trained-from-scratch simple supervised model.

**Weaknesses:**

Weaknesses:

- The supervised learning baselines are useful, but are rather weak.  For example, in the context of histone modifications, a model such as Sei which is trained on much larger datasets has achieved much higher accuracy (although the results are not strictly comparable).  I expect such a model to perform much better than the DNA LM approach.

- Regarding the enhancer prediction task:  This is a case where additional forms of data such as DNA accessibility and DNA contacts are able to provide additional hints that make this problem more tractable, and likely provide much higher accuracy than reported here (see e.g. the EPCOT model from the Liu lab).

- The gene-finding task is somewhat contrived, as the resulting prediction need post-processing is required to convert those predictions into a coherent gene model.  Also, it's not clear how alternative splicing was handled, as it makes it difficult to assign a strict label to each position.

**Questions:**

- In Table 3 the authors provide accuracy from the literature.  Not clear how relevant those numbers are since they are based on potentially very different datasets.  It should be noted in the manuscript that those numbers should be taken with a grain of salt.

- The authors note that "reasoning over very long contexts, as e.g. required for finding enhancers, is still challenging."  Is this too much to expect from these models?  (i.e expect them to reason over long sequence contexts). Perhaps we should be content with LMs producing representations that are based on short contexts, and leave long-range contexts for downstream models?

- The authors say the provide "An adaptable benchmarking framework for preparing embeddings and training lightweight supervised models."  This point requires some elaboration - I assume you are referring to the code in the (currently anonymous) github repository.

- You had noted that your benchmark's advantage is a mix of short and long sequence classification; however, I think you could have chosen better problems for long sequence tasks, e.g. gene expression prediction; also, the histone modification prediction task might have benefitted from longer sequence contexts.

Minor comments:

- A comment regarding splice site classification:  You note that "Moreover, there are cases in which a short-sequence task represents a simplification compared to real-world applications, as exemplified by SS-containing sequences. In genome annotation, classifying SSs is a subproblem of the gene finding task and would typically not be performed on its own."  This is mixing the issue of short sequences and how a splice site classifier would be used.  From my experience, SS classification is a relatively easy task that does not really need long sequences; while it is used in gene finding, it has other applications related to gene annotation; the condition-dependent version of the problem stands on its own (see e.g. the recent paper on the pangolin deep learning model).

- "The availability of unlabeled genomic sequences and limited labeled data appear to make language modeling a natural fit for DNA."
There is actually a wealth of labeled data - models like Sei and Enformer are very successful at leveraging large scale labeled data.

- "Gene-adjacent regulatory elements are referred to as cis, and distant ones as trans."  The definition of cis and trans is not a matter of distance!

- Reference missing in 3.3:  and follows the methodology of ??.

---

> ### Author Response · Authors · 2023-11-16
> **First reponse to reviewer vBTw (1/3)**
>
> We thank the reviewer for their detailed comments! We have updated our manuscript accordingly, improving the clarity of various aspects as outlined below, and are looking forward to further discussion.
> # Weaknesses
> > The supervised learning baselines are useful, but are rather weak. For example, in the context of histone modifications, a model such as Sei which is trained on (...)
>
> It is true that for many tasks, there are available methods that are trained with supervision on larger datasets and that they should yield higher accuracy by both architectural innovation as well as transfer learning from multiple datasets.
>
> However, a key objective of curating expert methods for BEND was that they could be evaluated on the same data on comparable terms as far as possible, so that LM performance can be put in context directly (see also answer below regarding literature methods in Table 3). For Sei, which was trained on ~22k tracks, it would be difficult to answer in direct comparison whether unsupervised DNA LM features are beneficial for predicting histone modifications in a given dataset. The Basset (Kelley 2017) model, which we trained on the exact same data as the CNN models operating on DNA LM features, allows us to compare performance directly - revealing that DNA LM features can be beneficial, but also that fully supervised approaches are very competitive when the data is large. We have updated the “Supervised baselines” section to better explain the role of the task-specific methods sourced from literature.
>
> > Regarding the enhancer prediction task: This is a case where additional forms of data such (...)
>
> We fully agree that experimental functional genomics data renders enhancer prediction a more tractable problem, and should in practice always be leveraged for developing state-of-the-art enhancer prediction methods. That being said, the focus of BEND is evaluating DNA LMs - and as such, we were interested in probing to what extent (unsupervised) DNA LMs encode features useful for this task, comparing against baseline approaches and existing methodology (Enformer).
>
> In fact, based on the observed results, we would absolutely advise against using currently available DNA LMs without additional inputs for this task in practice. We have extended the results discussion to highlight that our findings refer to enhancer annotation without using supporting experimental data as input.
>
> > The gene-finding task is somewhat contrived, as the resulting prediction need post-processing is required to convert those predictions into a coherent gene model. Also, it's not clear how alternative splicing was handled (...)
>
> This is correct - from a practical use perspective (where the aim is to report protein coding sequences, not delineate gene structure as nucleotide labels), the predicted labels would be post-processed and converted into a list of CDS (and other elements of interest). However, phrasing the prediction problem itself as a sequence-to-sequence task is a common approach that is also followed by HMM-based gene finders such as AUGUSTUS (Stanke and Waack, 2003). We note that our 9-class formulation is intentionally more simplistic compared to AUGUSTUS, which uses a state space that is highly tailored to the HMM framework.
>
> We think therefore that the difference is not how the prediction task is cast, but rather whether performance is computed on the “raw” seq2seq or the post-processed output. We opted to report performance directly on the nucleotide-wise predictions, as this
> 1) simplifies the task’s definition - we believe that the high intricacy of the state space models used by HMM-based gene finders is a key hurdle to its widespread adaptation as a benchmark task, even though it has a rich history in bioinformatics, and
> 2) allows us to compute detailed performance metrics for each class. This is useful for investigating in detail which elements of gene structure an LM’s embeddings capture. We have now added tables of these metrics to the supplement, revealing a large difference in how well the different LMs capture the highly important donor/acceptor sites and thus how well the embeddings capture the patterns of gene structure in the raw nucleotide sequence. Additionally, we have updated the supplement to describe in detail how AUGUSTUS is benchmarked on the dataset.
>
> In practice, when the aim is to develop competitive gene finders, rather than probing and comparing the capacity of different DNA LMs, we argue in favor of using more sophisticated decoders over a 2-layer CNN, which we discuss in our results section. Potentially, such decoders will operate more like existing HMMs to enforce gene model coherence explicitly.
>
> Lastly, we think that alternative splicing prediction would be unsuitable for establishing an easy-to-reuse benchmark task. Alternative splicing products were therefore excluded from the benchmark set. We have updated the gene finding datasheet to discuss this.

---

> ### Author Response · Authors · 2023-11-16
> **First reponse to reviewer vBTw (2/3)**
>
> # Questions
> > In Table 3 the authors provide accuracy from the literature. Not clear how relevant those numbers are since they are based on potentially very different datasets. It should be noted in the manuscript that those numbers should be taken with a grain of salt.
>
> We realize now that labeling this row as “Literature” was ambiguous. In fact, all performance metrics provided in the table were computed on BEND’s datasets and we can confirm that they are thus directly comparable. “Literature” was merely meant to convey that these are task-specific expert methods curated from existing literature.
>
> We have now renamed the row to “Expert method” to better communicate its meaning, and have added an explanation to section 4-Modeling.
>
> > The authors note that "reasoning over very long contexts, as e.g. required for finding enhancers, is still challenging." Is this too much to expect from these models? (...)
>
> Thank you for raising this interesting question! We tend to agree that it might be too much to expect self-supervised models to learn features that capture genomic organization over up to 50k bp by masked language modeling only. With recent DNA LMs increasingly claiming improved scaling to longer input lengths, we think it is crucial to establish long-range benchmark tasks to investigate quantitatively whether scaling the input length is really helping. Our results do not suggest that there is a direct effect of pretraining on longer ranges on long-range task performance.
>
> Should we therefore be content with what we have? This is hard to answer in general, as it very much depends on the task that needs to be solved. For many genomic problems, having long-range understanding would undoubtedly be beneficial. We hypothesize that the future might bring hybrid approaches, where masked language modeling allows a model to learn local context features, and readily available functional genomics data provides a supervised signal for learning long-range dependencies.
>
> > The authors say they provide "An adaptable benchmarking framework for preparing embeddings and training lightweight supervised models." This point requires some elaboration (...)
>
> This is correct - BEND is set up in a way that allows new tasks to be added in a straightforward manner (a bed file with coordinates, splits and labels,  and YAML-formatted configuration that defines the task). We also provide instructions for adding new LMs, and strive to keep the repo updated with regards to new DNA LMs (e.g. by having added the NT-V2 LMs since the original submission).
>
> The BEND repository has sphinx documentation that highlights how adding new embedders and tasks works in practice - the documentation source is included in the anonymized repository. We apologize that this is inconvenient to browse for review with the constraints imposed by anonymity, but will be hosted in proper HTML form eventually.
> > You had noted that your benchmark's advantage is a mix of short and long sequence classification; however, I think you could have chosen better problems for long sequence tasks, e.g. gene expression prediction; also, the histone modification prediction task might have benefitted from longer sequence contexts.
>
> This is a great point - tasks that are derived from genome-wide functional genomics experiments can technically be defined over *any* range (up to the limits imposed by chromosomes) - as seen in e.g. the recent Borzoi model (Linder 2023), that defined expression prediction over 524 kbp. Within the benchmark, where the defined aim is to investigate LMs and compare the utility of their pretrained features, we opted for a length that fits the context of all included DNA LMs, and have therefore chosen to predict histone modifications from 512bp, the context window of DNABERT. We absolutely agree that longer context should be beneficial to these tasks (provided that an LM learned to exploit the context).
>
> Establishing BEND’s data in bed-based format also makes it straightforward to change the flanking context at will, which we hope renders the materials more reusable and flexible for future LMs. We mention this in section 3, and have provided a more detailed discussion in section A.2.
>
> For BEND, we aimed at specifically establishing long-range tasks that have a natural biological context length imposed by the task - rather than the context length being driven by computational scaling constraints. Since very long context ranges are common in the genome and its regulation, we believe it is vital for the future utility of DNA LMs to benchmark and aim for good performance on tasks that take such ranges into account.

---

> ### Author Response · Authors · 2023-11-16
> **First reponse to reviewer vBTw (3/3)**
>
> # Minor comments
> > A comment regarding splice site classification: You note that "Moreover, there are cases in which a short-sequence task represents a simplification compared to real-world applications, as exemplified by SS-containing sequences. In genome annotation, classifying (...)
>
> Thanks for this comment. Indeed, splice site prediction has applications beyond the context of gene finding - we have now rewritten this section to better express the original idea (SS prediction performance being insufficient to reason about gene finding performance) without unduly claiming a limited scope of SS prediction on its own.
>
> > "The availability of unlabeled genomic sequences and limited labeled data appear to make language modeling a natural fit for DNA." There is actually a wealth of labeled data - models like Sei and Enformer are very successful at leveraging large scale labeled data.
>
> We have now rewritten this statement - it is true that there is in fact plenty of labeled data available compared to most other areas of biology. Our intent was to express that we, for the time being, have more genomes than we have functional genomics experimental coverage - which would make self-supervised representation learning very attractive for cases where functional genomics data is not available.
>
> > "Gene-adjacent regulatory elements are referred to as cis, and distant ones as trans." The definition of cis and trans is not a matter of distance!
>
> Thank you for bringing this to our attention - we have removed this erroneous statement.

---

### Official Review · Reviewer_cZPk · 2023-10-30

**Soundness:** 4 excellent
**Presentation:** 4 excellent
**Contribution:** 2 fair
**Rating:** 6
**Confidence:** 5

**Summary:**

This paper presents a collection of benchmark tasks that are intended to measure the performance of a DNA language model.  The tasks are quite diverse, both in terms of the phenomena they cover and their difficulty level.  Providing such a benchmark is useful, though several recent studies have already done this. The main novel contribution here is including tasks that take very long sequences as input, such as predicting enhancer-promoter interactions.

N.B. I increased my score by one point after reading the reviewer response.

**Strengths:**

I really liked the section that provided biological background -- it was clear and concise.

The benchmark tasks are well described, and each one is important for a DNA language model to be able to address.

Some of these tasks are more challenging than the ones used in previous studies.

**Weaknesses:**

Including only tasks from the human genome is problematic.  It seems clear that a good language model of DNA should cover more than just one species.  The most recent competing benchmark (Gresova 2023) includes eight tasks from three different species.

This benchmark does not improve very much over the Gresova benchmark published this year.

minor:

It would be good to point out, in Section 2.1, that these descriptions are about eukaryotic genomes.

Introduce "secondary structure" before using the abbreviation "SS."

**Questions:**

Why are enhancers defined to be 128bp?  I think of an enhancer as minimally corresponding to a missing nucleosome, plus its linker, which would be 225-250bp.  Some are significantly larger.

Why did you choose to frame the enhancer task as annotation rather than matching?  It seems weird to refer to this as detecting "long-range interactions," since the task does not require that a given enhancer actually be operating on the gene at the center of the selected window.  I also don't understand in what sense this annotation framing of the problem can be considered "more stringent."

Why did you choose not to do fine-tuning for each task?  It's not clear to me that, in practice, anyone would adopt a fully unsupervised approach to this kind of supervised problem.

---

> ### Author Response · Authors · 2023-11-16
> **First reponse to reviewer cZPk (1/2)**
>
> Thank you for your helpful feedback! We have improved our manuscript, clarifying various aspects, as outlined below. We welcome further discussion.
>
> # Weaknesses
> > Including only tasks from the human genome is problematic. It seems clear that a good language model of DNA should cover more than just one species. The most recent competing benchmark (Gresova 2023) includes eight tasks from three different species.
>
> We fully agree that evaluating on multiple species is a desirable feature of a DNA LM benchmark. Our reason to focus on the human genome is twofold:
> A) only evaluating multi-species DNA LMs would have reduced the scope of DNA LMs that could be benchmarked to a single Nucleotide Transformer model and DNABERT2.
> B) A key problem with Gresova 2023 is that tasks are not matched between different species. Ultimately, a benchmark should allow us to disentangle understanding of a specific task, and understanding of specific organisms. This would require a task to be defined equally over many different organisms - in Gresova 2023, that is unfortunately not yet the case:
>
> - 3/4 of Gresova et al.’s non-dummy/demo tasks are defined on human only (ensembl_regulatory, ensembl_ocr, nontata_promoters)
> - The enhancer tasks (human_enhancers_cohn, human_enhancers_ensembl, drosophila_enhancers_stark) are very heterogeneous in how they are defined and not directly comparable between human and drosophila (evidence of the underlying data, length of samples). We would argue that in a true multi-species benchmark, the task’s format and objective should be harmonized.
> - Lastly, the organisms roundworm (demo_human_or_worm) and mouse (dummy_mouse_enhancers_ensembl) are only included in a  “demo” (computationally generated negatives) or “dummy” (meant for prototyping, not benchmarking) task. Moreover, the roundworm task does not relate directly to any specific genomic feature.
>
> We see the establishment of a true multi-species benchmark covering diverse organisms as a key future objective that is highly dependent on the availability of experimental data of a comparable quality to the data underlying BEND, and Gresova et al.’s work highlights some of the challenges precluding its establishment. We address this in our Limitations and Outlook section. We have also updated our related works section to discuss the species coverage of Gresova 2023.
>
> > This benchmark does not improve very much over the Gresova benchmark published this year.
>
> While we have not focused the presentation of our manuscript on the relation to Gresova et al. extensively, we would like to highlight the key improvements of BEND:
> - Experiments: DNA LMs were benchmarked comprehensively, providing quantitative insights into the effectiveness of self-supervised models. Gresova et al. does not include any DNA LMs and does not include any experimental setup for benchmarking them.
> - Experiments: Task-specific expert model performances were established. These allow us to directly put any attained DNA LM or simple baseline performance into context of a tailor-made task specific model that sees widespread use.
> - Benchmark: Interpretability - each task covers distinct phenomena of genomic organization and is biologically interpretable. Moreover, the tasks aim at staying faithful to imbalance that is intrinsic to the genome, without taking steps to artificially rebalance label ratios.
> - Benchmark: Task length - BEND’s task cover a length range up to 50kbp
> - Benchmark: Dataset scale - leveraging functional genomics data, we established datasets reaching up to 2 million samples.
>
> In summary, while we consider Gresova et al 2023 a relevant resource of balanced classification tasks that is useful for evaluating and developing models operating on DNA, we are convinced that BEND is ultimately better suited for gaining insights into the performance of DNA LMs.
>
> > minor:
> It would be good to point out, in Section 2.1, that these descriptions are about eukaryotic genomes.
>
> > Introduce "secondary structure" before using the abbreviation "SS."
>
> We have now ensured that this is properly addressed in the section and changed the title to “Eukaryotic DNA organization and terminology”. The abbreviation “SS” for splice site is now also properly introduced before its first usage.

---

> ### Author Response · Authors · 2023-11-16
> **First reponse to reviewer cZPk (2/2)**
>
> # Questions
> > Why are enhancers defined to be 128bp? I think of an enhancer as minimally corresponding to a missing nucleosome, plus its linker, which would be 225-250bp. Some are significantly larger.
>
> Indeed, enhancers are larger than 128bp and we need to describe our task definition more clearly. In our enhancer annotation task, multiple consecutive 128-bp bins are labeled as “enhancer” if they span an enhancer element. The binning is motivated by the fact that experimental evidence does not necessarily resolve enhancer boundaries at single-nucleotide resolution. It also matches the formulation of enhancer annotation in Enformer (Avsec et al, 2021), which we compare to.
>
> We have now rephrased the statement in the data section to better express that all bins that are part of a (longer-than-128bp) enhancer are labeled. We have also added a figure of the enhancer elements length distribution in the appendix.
>
> > Why did you choose to frame the enhancer task as annotation rather than matching? It seems weird to refer to this as detecting "long-range interactions," since the task does not require that a given enhancer actually be operating on the gene at the center of the selected window. I also don't understand in what sense this annotation framing of the problem can be considered "more stringent."
>
> It is true that in general, enhancers can be anywhere, and operate on genes that are not adjacent to them. This is why in our enhancer annotation task, all included enhancers were experimentally validated to operate on the gene of interest that the window is based on. The selection of enhancers follow the same criteria used to evaluate unsupervised enhancer discovery in Enformer (Avsec et al, 2021). The task is thus gene-centric: Given a gene, annotate enhancers in a 100kb window that operate on this gene. We have updated the datasheets section to better explain the nature of the experimental data.
>
> “More stringent” was meant to express that finding enhancers in a 100kbp window is presumably more challenging than e.g. classifying a balanced set of enhancer/non-enhancer sites for a gene. We have now removed this sentence as it is not necessary to explain the task.
>
> > Why did you choose not to do fine-tuning for each task? It's not clear to me that, in practice, anyone would adopt a fully unsupervised approach to this kind of supervised problem.
>
> The aim of BEND is to evaluate the utility of pretrained features for biologically relevant downstream tasks, understanding what genomic features different DNA LMs learn from self-supervision. We consider fine-tuning not strictly necessary to investigate this research question, and train lightweight supervised CNN models that operate on the LM’s embeddings as inputs. This paradigm (keeping LMs frozen) has also seen widespread use in the adjacent field of protein LMs, and has the advantage that it scales to any model size or architecture directly, without further considerations on how to effectively fine-tune models of different architectures or encountering computational constraints.
> We very much agree that it would be unreasonable to take a fully unsupervised approach in practice - hence the CNN is trained with supervision. When the aim is to push performance further, rather than probing the utility of LM embeddings, more sophisticated downstream models and/or fine-tuning should be considered, which we argue in favour of in our discussion.

---

### Official Review · Reviewer_hBHB · 2023-11-01

**Soundness:** 2 fair
**Presentation:** 3 good
**Contribution:** 2 fair
**Rating:** 5
**Confidence:** 5

**Summary:**

Summary of the paper:
The paper presents BEND, a benchmark for DNA language models that focuses on biologically meaningful tasks defined on the human genome. The aim is to evaluate the ability of unsupervised language modeling techniques in annotating various functional and regulatory elements within DNA sequences. The results show that while current DNA LMs can rival expert methods in some tasks, they struggle with capturing long-range features in the DNA.

**Strengths:**

- The overarching motivation behind creating BEND, which emphasizes understanding the genome across longer ranges, is commendable.
- The efforts to collect the benchmark dataset are commendable, and the dedication shown in running benchmarks for so many language models and supervised baselines is admirable.

**Weaknesses:**

- However, I feel that the task formulation for long sequences lacks depth and isn't entirely persuasive.
- When benchmarking DNA Language models, it's crucial to explore the intricacies of training at least one of these models from scratch. This would provide a comprehensive insight into their potential and the limitations of pretraining.
- A side-by-side comparison with a DNA Language model trained from scratch is essential. Such an analysis would give a more rounded perspective on the strengths and shortcomings of the existing methods.

**Questions:**

I believe this work has significant potential. If the authors address the concerns raised in my comments, I would be inclined to recommend a higher score for this submission.

- Figure 1 appears to inaccurately represent the length of certain genomic features. For example, the majority of exons in the human genome are shorter than 200 base pairs, with an average length between $120-170$ base pairs. Furthermore, promoter lengths usually range from 100 to 1000 base pairs. The diagram should at least offer a rough indication of these lengths. It's also crucial to highlight that exons and introns alternate in their appearance, a detail that the figure should encompass.
- In the introduction, l'd suggest mentioning foundational works like DeepSEA alongside DeepBind when discussing supervised learning on DNA. These groundbreaking studies have left an indelible mark on the field.
- Concerning the enhancer annotation, there is some uncertainty regarding the authenticity of the ground truth. The ABC method is based on inference and might not reflect a direct experimental outcome.
- Clarity is sought on the dataset splitting methodology - why was the data from the pretraining phase included? What about considering a leave-chromosome-out approach to splitting? The methods to prevent data leakage should also be elucidated.
- The decision to categorize variant effect prediction as a binary problem raises questions. Given the intricate nature of genomic variations and the subsequent implications, framing this as a regression task might be more appropriate.
- It's crucial, especially with such intricate datasets, that downstream models be evaluated using a leave-chromosome-out strategy to guarantee robust results.
- Te absence of a simple CNN baseline for the variant effect prediction task is a notable omission. Such a baseline would not only validate the implemented supervised models but also provide an insight into the relative performance of more complex models.

---

> ### Author Response · Authors · 2023-11-16
> **First response to review 1/2**
>
> We thank the reviewer for their comprehensive feedback and are happy to hear that they see significant potential for BEND! We have updated the manuscript according to the feedback, as outlined below, and look forward to further discussion.
>
>
> # Weaknesses:
> > However, I feel that the task formulation for long sequences lacks depth and isn't entirely persuasive.
>
> Including feedback from all reviewers, we have now clarified various aspects of the underlying experimental data of the two long-range tasks, and extended our description of how their formulation follows existing task-specific work (AUGUSTUS Stanke & Waack 2003, Enformer Avsec et al. 2021). Formulating tasks in a way that enables them to be evaluated by previous task-specific methods was of key interest when establishing BEND. We would be happy to address any further comments regarding their persuasiveness.
>
> > When benchmarking DNA Language models, it's crucial to explore the intricacies of training at least one of these models from scratch. This would provide a comprehensive insight into their potential and the limitations of pretraining.
>
> > A side-by-side comparison with a DNA Language model trained from scratch is essential. Such an analysis would give a more rounded perspective on the strengths and shortcomings of the existing methods.
>
> Indeed, we were interested in training LMs from scratch. We trained both a dilated CNN as well as an AWD-LSTM DNA LM that we included in all experiments. The dilated CNN was motivated by the fact that this architecture had shown good performance in GPN on A. thaliana (Benegas et al. 2023), whereas the AWD-LSTM was considered a very simple, but computationally cheap model to put gains of more expensive architectures into context.
> As suggested, having this comparison does give a more rounded perspective: On many tasks, the dilated CNN shows competitive performance with medium-sized transformer-based DNA LMs. That being said, the variety of models provided by Nucleotide transformer allows us to reason about pretraining intricacies, even when models of the scale of NT-MS are out of reach to reproduce in this work (reported: 128 A100s for 28 days). The comparison of NT-V2 (now added to the manuscript) and NT-MS to the other NT-family models reveals that a multispecies pretraining dataset can yield better understanding of gene structure unique among all investigated models. To our best knowledge, this was not reported before.
>
> # Questions
> > Figure 1 appears to inaccurately represent the length of certain genomic features. For example, the majority (...)
>
> Thank you for the feedback, we have updated the figure. The figure now has updated lengths, and we have added an additional exon and intron to highlight their alternation. We have also extended the figure’s caption accordingly.
>
> > In the introduction, l'd suggest mentioning foundational works like DeepSEA alongside DeepBind when discussing supervised learning on DNA. These groundbreaking studies have left an indelible mark on the field.
>
> Thank you for the suggestion - We have now ensured that DeepSEA is mentioned at the same time as DeepBind.
>
> > Concerning the enhancer annotation, there is some uncertainty regarding the authenticity of the ground truth. The ABC method is based on inference and might not reflect a direct experimental outcome.
>
> It is correct that the ABC method in itself is a predictive method and should not be considered ground truth. We want to avoid uncertainty in the labels of the benchmark datasets and therefore did not consider ABC-predicted enhancers for BEND. Rather, all included enhancers were validated directly using either CRISPRi-FlowFISH (Fulco et al 2019) or crisprQTL (Gasperini et al 2019).
> We have updated the datasheet in the supplement to explicitly state that ABC-predicted enhancers are not part of the dataset.

---

> ### Author Response · Authors · 2023-11-16
> **First response to review 2/2**
>
> > Clarity is sought on the dataset splitting methodology - why was the data from the pretraining phase included? What about considering a leave-chromosome-out approach to splitting? The methods to prevent data leakage should also be elucidated.
>
> Having adequate splits for reliably quantifying performance is of key interest to us. As outlined in our answer further down below, we have ensured that individual datasets are split according to applicable best practices, either using a leave-chromosome-out approach, or splitting based on sequence homology for the gene finding task. The approach is now better highlighted in the text. These steps should largely prevent data leakage between partitions.
> Regarding data from the pre-training phase, following established practices in the protein language modeling field, we do not consider input sequence overlap between unlabeled pre-training data and labeled task data as leakage. We understand that data leakage from pre-training can be an issue when evaluating e.g. LLMs on text, where it is possible that the answer (=label) of a data point is encountered in conjunction with the data point during pre-training. When pre-training on unlabeled genomes, followed by evaluation on classification tasks, this naturally cannot occur. We therefore consider our approach aligned with established best practices and are confident that leakage risks are minimized.
>
> > The decision to categorize variant effect prediction as a binary problem raises questions. Given the intricate nature of genomic variations and the subsequent implications, framing this as a regression task might be more appropriate.
>
> Indeed, variant effect prediction is a problem that allows for many different formulations, and for BEND, we opted to frame it as a classification task as done previously in DeepSEA and Nucleotide Transformer. Ultimately, the decision whether a task is categorized as classification or regression is driven by the available data. The underlying data that we used (eQTL+background SNPs, pathogenic+benign SNPs) provides categorical labels, making it a natural choice to render it as a classification problem.
> We fully agree that variant effect regression, as often seen in the protein LM field (e.g. Meier et al 2021), would be a valuable complementary task - we are however not aware of any experimental data on the human genome that would be comparable. That being said, we find that the formulation as a binary task allows us to gain insights into what features LMs learn - as exemplified by NT-MS’ unique understanding of splice site variants.
>
> > It's crucial, especially with such intricate datasets, that downstream models be evaluated using a leave-chromosome-out strategy to guarantee robust results.
>
> Thank you for raising this important point. We agree that this is important and we should have clarified our splitting strategy better. We do actually split by chromosome for all tasks except for gene finding (where sequence homology-based splitting based on the coding sequence is more appropriate and commonly used). We have now ensured that the splitting strategy is described in detail in the supplementary datasheet for each task, and added a statement in section 3-Tasks and Datasets explaining that we follow a chromosome-based splitting approach in general.
>
> > The absence of a simple CNN baseline for the variant effect prediction task is a notable omission. Such a baseline would not only validate the implemented supervised models but also provide an insight into the relative performance of more complex models.
>
> Thank you for this suggestion, we should clarify why there is no CNN baseline for variant effect prediction. This task was rendered as a zero-shot classification task (following Dalla-Torre et al 2023), using the LM’s embedding cosine distance as the predictor. Therefore, no supervised models were trained on the task, and a supervised shallow CNN baseline would not be applicable for comparison.  We have now updated section 4-Modeling to discuss this accordingly.

---

### Author Response · Authors · 2023-11-16
**Response to all reviewers**

We thank all the reviewers for their constructive feedback! We are excited to hear that the reviewers consider the established tasks *important and more challenging* (cZPk),  *more realistic* (vBTw) and *relevant to modern biology* (gFHh). We have uploaded a revised version of the manuscript addressing the comments, and responded to reviewer comments individually point-by-point.

In addition to the individual points, we have made the following improvements to the manuscript:

* We have added an **additional task** on **chromatin accessibility prediction**.
* The related works section was updated to consider the **latest update** of the **Nucleotide Transformer preprint**.
* Experimental results now include the **Nucleotide Transformer V2** model with extended input length.
* We extended the supplementary material with more **detailed per-label performance for all tasks** to provide further insights into LM performance.

Overall, we consider BEND an adequate manuscript for ICLR, as we believe that our contribution serves to help advance an emerging field of relevance to machine learning in genomics. In line with previous influential benchmark works published at this venue, such as GeneDisco (Mehrjou, ICLR ‘22) and Long Range Arena (Tay, ICLR ‘21), BEND contributes a collection of original datasets and experiments.

We consider it particularly important to faithfully cover relevant research questions in order to help guide the creation of useful foundation models. As such, tasks were curated with a focus on experimental data quality, realism and biological relevance. Our experiments offer novel insights into DNA LMs, such as a unique ability of models pre-trained on multi-species data to capture gene structure. Finally, we stress the importance of establishing a benchmark that is developed independently from any specific language model - such that we guarantee that it carries neither deliberate nor unintentional bias towards any existing model.

---

### Author Response · Authors · 2023-11-23

Dear Reviewers and ACs,

As the rebuttal and discussion period is very nearly over we're are and we have not heard any response to our rebuttal, we still hope that you will take it into account going forward in the process. We have made an effort to address all concerns of the reviewers and clarify any misunderstandings. We believe that their suggestions have improved the paper.

For a briefer summary of our rebuttal this [comment](https://openreview.net/forum?id=uKB4cFNQFg&noteId=Mt6NnREQjm).

We thank you for your reviews.

---

### Meta-Review · Program_Chairs · 2024-01-15

**Metareview:**

The paper presents a benchmark for DNA language models that focuses on biologically meaningful tasks defined on the human genome. The main conclusion is that current DNA LMs find it difficult to capture long-range features in DNA, even as they exhibit expert-like performance on some tasks.

The reviews of the paper were mixed. The reviewers found the overall motivation strong but raised some concerns about experimental baselines and discussion of related work. However, the authors have correctly pointed out that some of these concerns are unfounded and have also updated the draft to address some of the others. Overall, based on my reading of the paper and the reviews, I consider the paper above the bar as a datasets-and-benchmarks paper.

**Justification For Why Not Higher Score:**

While the paper represents a solid contribution, it's not outstanding in terms of originality and likely impact.

**Justification For Why Not Lower Score:**

The paper represents a solid contribution that is likely to aid progress in an emerging area.

---

### Decision · Program_Chairs · 2024-01-16

Accept (poster)